# Simple yet Effective: Low-Rank Spatial Attention for Neural Operators

**Zherui Yang**[1]  **Haiyang Xin**[1]  **Tao Du**[2,3]  **Ligang Liu**[1]

## Abstract

Neural operators have emerged as data-driven surrogates for solving partial differential equations (PDEs), yet capturing the long-range spatial dependencies inherent in physical systems remains computationally expensive when high accuracy is required. Across many PDE regimes, the underlying global kernels exhibit rapid spectral decay, implying that the dense interaction map is amenable to a low-rank factorization. We leverage this observation to give a unified view of global mixing approaches in representative neural operators as instantiations of a common low-rank template. Guided by this view, we introduce Low-Rank Spatial Attention (LRSA), a direct instantiation of the template that compresses point features into a small set of latent tokens, performs global mixing in the latent space, and reconstructs context back to spatial points. Unlike prior latent-token approaches that rely on non-standard aggregation or normalization modules, LRSA is built purely with standard Transformer primitives, ensuring architectural simplicity and facilitating the use of hardware-optimized kernels, enabling robust mixed-precision training. In our experiments, such a simple construction is sufficient to achieve high accuracy, yielding an average error reduction of over 17% relative to second-best methods, while remaining stable and efficient in mixed-precision training.

## 1. Introduction

Solving partial differential equations (PDEs) is fundamental for modeling physical and engineering systems, with applications ranging from weather forecasting to fluid dynamics and structural analysis (Bonev et al., 2023; Horie & Mitsume, 2024; Li et al., 2022b). Traditional numerical solvers (e.g., finite element methods) discretize the domain and solve large algebraic systems, whose computational costs grow rapidly with the number of degrees of freedom as resolution increases. Recently, learning-based surrogates, in particular neural operators (Kovachki et al., 2023; Lu et al., 2019; Azizzadenesheli et al., 2024), have emerged as a promising alternative by learning mappings between function spaces and enabling fast inference after training.

A central challenge in neural operator design is to capture the long-range, global dependencies inherent to physical phenomena, while keeping computation and memory costs scalable with the number of spatial points. The community has explored several directions. Among them, spectral and basis-based operators perform global mixing through a compact set of modes. For example, Fourier Neural Operator (FNO) and its variants (Li et al., 2020b; 2022b; Kossaifi et al., 2023) employ Fourier bases with high-frequency truncation to achieve global mixing. On irregular geometries, Fourier bases are replaced by Laplace eigenfunctions to obtain geometry-aware global representations (Yue et al., 2025; Chen et al., 2024). While effective, these methods fundamentally rely on predefined analytical bases, which often necessitates specific discretizations or geometric priors to function correctly.

Attention mechanisms (Vaswani et al., 2017) provide a flexible, data-driven way to model global correlations, but full attention scales quadratically with the number of points, which is prohibitive for high-resolution PDE prediction. To reduce this cost, linear-attention variants reduce complexity by reordering computations or approximating the attention kernel (Cao, 2021; Li et al., 2022a), and structured designs further exploit separable patterns in spatial interactions (Li et al., 2023b). A complementary direction introduces an explicit latent bottleneck. Transolver and follow-up work (Wu et al., 2024; Hu et al., 2025) learn a data-driven compression by explicitly slicing and aggregating point-wise features into a fixed number of latent tokens, enabling accurate global mixing with near-linear scaling when the latent size is fixed. In practice, however, maintaining well-conditioned slice assignments can require additional normalization and auxiliary mechanisms to mitigate slice degeneration (Luo et al., 2025), moving the design away from a purely standard attention pipeline.

[1]University of Science and Technology of China, Hefei, China [2]Tsinghua University, Beijing, China [3]Shanghai Qi Zhi Institute, Beijing, China. Correspondence to: Ligang Liu <lgliu@ustc.edu.cn>.

*Proceedings of the 43rd International Conference on Machine Learning*, Seoul, South Korea. PMLR 306, 2026. Copyright 2026 by the author(s).

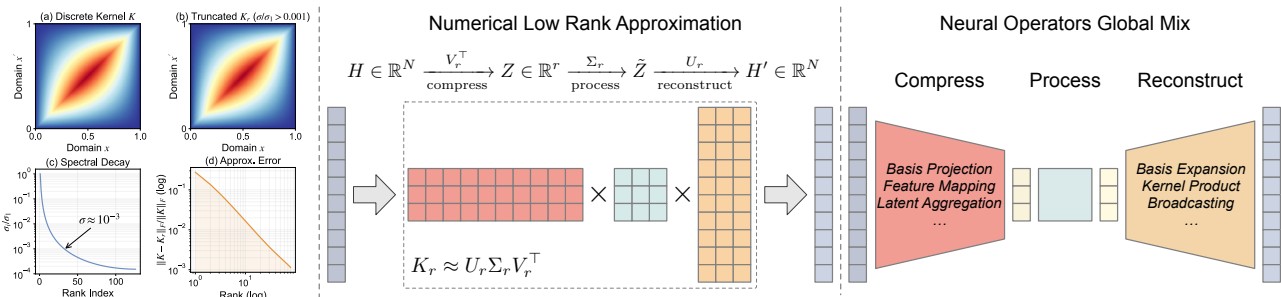

*Figure 1.* **Compressibility of PDE interactions and the unified low-rank paradigm. Left (1D Poisson):** (a) original dense kernel, i.e., the Green function; (b–d) underlying low-rank properties. reconstructed kernel, fast spectral decay, and approximation error—derived from the numerical factorization illustrated in the middle. **Middle:** numerical low-rank approximation of global interactions via $K_r \approx U_r \Sigma_r V_r^\top$. **Right:** diverse neural-operator global-mixing modules unified as a learnable *compress–process–reconstruct* template.

Notably, an intrinsic low-rank structure unifies both the physical nature of PDE interactions and the architectural design of efficient neural operators. From a physics perspective, dense PDE interaction kernels (e.g., Green's functions) often exhibit rapidly decaying singular values, making them well approximated by low-rank factorizations (Fig. 1, Left/Middle). From a modeling perspective, many operator backbones implicitly mirror this structure (Fig. 1, Right): they either utilize truncated spectral bases, factorize the dense attention kernel via feature maps (as in linear attention), or route information through a compact set of latent tokens. This dual observation motivates a unified view of global mixing as a *compress–process–reconstruct* factorization. However, realizing this structure often imposes constraints, such as prescribed bases or symmetries, which can complicate implementation and hinder the direct use of hardware-optimized kernels (Dao et al., 2022). This raises a natural question: *Can we realize such a low-rank global mixing using only standard Transformer primitives, without relying on specialized components?*

In this work, we propose **Low-Rank Spatial Attention (LRSA)** as a direct answer. Restricting our design space strictly to standard Transformer primitives, i.e., scaled dot-product attention, normalization, and feed-forward networks (FFN), we derive a minimalist architecture that mirrors the structure of low-rank factorization. LRSA first compresses global information from spatial point features into a compact set of latent tokens via cross-attention between learnable latent queries and the point features, forming a data-driven basis that adapts across discretizations and geometries. It then processes global information within this bottleneck via standard self-attention and FFNs, mixing globally aggregated latent tokens to model couplings in the compressed subspace. Finally, a second cross-attention reconstructs spatial point features from the latent bottleneck, decoupling the reconstruction basis from compression for enhanced flexibility. Despite its simplicity, LRSA attains competitive accuracy across diverse PDE benchmarks while remaining

robust and efficient under mixed-precision training. Our contributions are summarized as follows:

- We unify global mixing in PDE operator learning under a low-rank perspective, interpreting several representative neural operators within this shared template.

- Building on this view, we propose Low-Rank Spatial Attention (LRSA), a minimal block instantiated purely with standard Transformer primitives, which makes the design amenable to optimized fused kernels.

- We validate LRSA on diverse PDE problems, demonstrating its improved accuracy over existing baselines, together with superior training stability and efficiency under mixed-precision regimes.

## 2. Related Work

We position LRSA within the broader landscape of neural operators, specifically focusing on spectral and basis-based methods and scalable attention mechanisms for modeling global physical dependencies.

**Spectral and Basis-Based Neural Operators.** A prominent paradigm in operator learning approximates solution operators via spectral transformations or basis expansions. Fourier Neural Operators (FNO) (Li et al., 2020b) and follow-ups (Kossaifi et al., 2023; Wen et al., 2022; Tran et al., 2023) achieve efficient global mixing by truncating frequencies and using FFT-based convolutions, leading to strong accuracy and favorable scaling on structured grids. To extend these ideas beyond regular grids, recent works incorporate geometric information through learned deformations (Li et al., 2022b), direct spectral evaluations (Lingsch et al., 2024), and hybrid GNN-spectral designs such as GINO (Li et al., 2023c). Conceptually, however, these methods still rely on a fixed global basis (e.g., Fourier modes) to capture long-range dependencies. Methods tailored to irregular domains, such as NORM (Chen et al., 2024) and HPM (Yue

et al., 2025), replace Fourier modes with eigenfunctions of the Laplace–Beltrami operator to obtain geometry-aware spectral bases. By contrast, we use learnable bases, enabling an adaptive, data-driven representation for irregular, mesh-free geometries.

**Efficient Attention for Operator Learning.** While standard Transformers (Vaswani et al., 2017) offer a flexible alternative for global modeling, their quadratic complexity poses a bottleneck for dense physical fields. To mitigate this, Galerkin Transformers (Cao, 2021) and Linear Attention variants (Katharopoulos et al., 2020; Li et al., 2022a) reduce complexity by reordering matrix multiplications or approximating the attention kernel with explicit feature maps. Other designs exploit structural assumptions for efficiency, such as axial factorization in Fact-Former (Li et al., 2023b). While these methods improve scalability, their approximation strategies can weaken the sharp, non-linear selectivity of the full softmax attention mechanism (Qin et al., 2022). In contrast, LRSA routes global mixing through a compact latent bottleneck, retaining the expressiveness of standard attention while maintaining near-linear complexity.

**Latent-Token and Bottleneck Architectures.** Our work is most closely aligned with architectures that decouple computational cost from input resolution via latent bottlenecks. In computer vision, Perceiver IO (Jaegle et al., 2021) pioneered the use of a fixed latent array to scale attention to massive multimodal inputs. In scientific machine learning, Universal Physics Transformers (UPT) (Alkin et al., 2024) extend this latent-backbone paradigm to PDE modeling. Unlike Perceiver-style architectures such as UPT, where the latent tokens constitute the backbone representation across layers, LRSA treats latents as an internal low-rank routing mechanism for global coupling within each layer. This design preserves the neural-operator convention that the spatial points are the primary state on which local channel mixing and supervision are defined. Transolver (Wu et al., 2024) aggregates spatial features into slices and broadcasts information back to points, achieving near-linear complexity when the number of slices is fixed. Subsequent works such as Transolver++ (Luo et al., 2025) introduce stochastic sampling mechanisms (e.g., Gumbel-Softmax) to refine the assignments, while LinearNO (Hu et al., 2025) simplifies the module by reinterpreting the aggregation step as a linear attention kernel. More recently, Transolver-3 (Zhou et al., 2026) extends this line of work to industrial-scale geometries through memory-efficient physics-attention reordering and decoupled inference with cached physical states. However, these approaches often rely on heuristic slice definitions or tightly couple the projection and reconstruction weights. In contrast, LRSA is built entirely from standard Transformer primitives, simplifying implementation and enabling hardware-optimized kernels.

# 3. Method

## 3.1. Problem Setup

We study learning PDE solution operators on a bounded domain $\Omega \subset \mathbb{R}^{d_{\text{phys}}}$. Inputs are given as an unordered point set $\{(x_i, f_i)\}_{i=1}^N$, with coordinates $x_i$ and features $f_i$. We predict target values $\{u_i\}_{i=1}^N$ on the same points (or query points), without assuming grid or mesh connectivity.

To process points, we lift coordinates and inputs into a $d$-dimensional feature space. Let $\text{PE}(\cdot)$ be a positional encoding (e.g., Fourier features) and $\text{Lift}(\cdot)$ be a point-wise feed-forward network (FFN):

$$
\begin{aligned}
h_i^{(0)} &= \text{Lift}\big([f_i, \text{PE}(x_i)]\big) \in \mathbb{R}^d, \\
H^{(0)} &= [h_1^{(0)}, \dots, h_N^{(0)}]^\top \in \mathbb{R}^{N \times d}.
\end{aligned}
\tag{1}
$$

We then apply $L$ pre-norm blocks to update point features, separating global spatial coupling from point-wise channel mixing (via FFNs). Concretely, each block takes the form

$$
\begin{aligned}
G^{(\ell)} &= H^{(\ell)} + \text{GlobalMix}\Big(\text{Norm}(H^{(\ell)}), X\Big), \\
H^{(\ell+1)} &= G^{(\ell)} + \text{FFN}\Big(\text{Norm}(G^{(\ell)})\Big),
\end{aligned}
\tag{2}
$$

where $X = [x_1, \dots, x_N]^\top$ and Norm denotes point-wise normalization, such as LayerNorm or RMSNorm (Ba et al., 2016; Jiang et al., 2023). We use GlobalMix to abstract the global coupling in neural operators. Since many PDE operators admit the integral form

$$
(\mathcal{K}h)(x) = \int_\Omega \kappa(x, y)\, h(y)\, dy,
$$

GlobalMix$(\cdot)$ serves as its discretization on sample points.

## 3.2. Compressibility of Global Interaction Maps

For a given layer, we associate GlobalMix with an induced interaction map $K(H, X) \in \mathbb{R}^{N \times N}$ such that

$$
\text{GlobalMix}(H, X) \approx K(H, X)\, H,
\tag{3}
$$

with $K(H, X)$ typically dense, representing all-to-all interactions among spatial points, and potentially conditioned on both the current features $H$ and the coordinates $X$. Materializing or applying $K$ explicitly leads to $\mathcal{O}(N^2)$ complexity, which is prohibitive at high resolution. Empirically, the dense interaction maps induced by PDE operators often exhibit fast spectral decay, aligning with classical compression results for integral operators (Bebendorf, 2008). This suggests that global coupling can be mediated through a small set of latent modes. Accordingly, we parameterize GlobalMix using the low-rank factorization

$$
K \approx UGV^\top, \quad U, V \in \mathbb{R}^{N \times M}, \; G \in \mathbb{R}^{M \times M}, \; M \ll N.
\tag{4}
$$

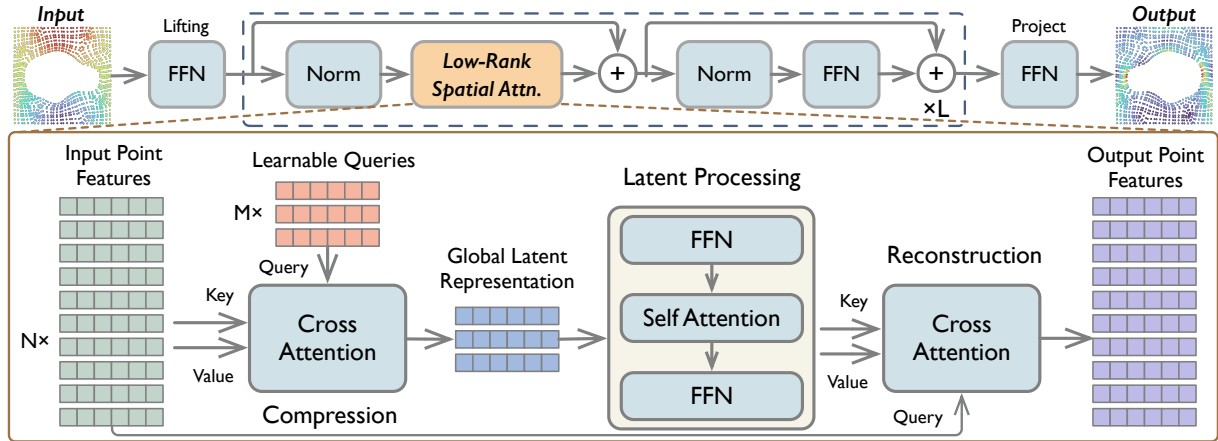

**Figure 2.** Overview of the neural operator backbone and the Low-Rank Spatial Attention (LRSA) block. LRSA routes global information through a compact latent bottleneck using only standard Transformer primitives.

Unlike standard SVD, we do not require orthogonality of $U, V$, nor do we assume $G$ is diagonal. Instead, we view Eq. (4) as a structural template: (i) **Compression** ($V^\top$) maps high-dimensional point features to $M$ compact latent tokens; (ii) **Processing** ($G$) mixes global information within this efficient latent space; and (iii) **Reconstruction** ($U$) broadcasts the refined global context back to the spatial domain. Next, we use Eq. (4) to make explicit the choices of compression/reconstruction ($U, V$) and latent mixing ($G$) in representative neural operators, thereby clarifying the design space targeted by LRSA.

### 3.3. A Unified Low-Rank View of Global Mixing

**Spectral / Basis-Based Operators: Fixed Bases** Let $\Phi \in \mathbb{R}^{N \times M}$ be the evaluation matrix of $M$ basis functions on the sampled points. A generic layer is

$$H' = \Phi G \Phi^\top H, \quad (5)$$

which is a low-rank GlobalMix with compression and reconstruction *fixed by a prescribed basis*, i.e., $U = V = \Phi$. FNO (Li et al., 2020b) corresponds to choosing $\Phi$ as truncated low-frequency Fourier modes on regular grids (with FFT-based compression/reconstruction). On irregular geometries, NORM (Chen et al., 2024) and HPM (Yue et al., 2025) replace Fourier modes with Laplace–Beltrami eigenfunctions, yielding geometry-aware but still fixed compression and reconstruction once the discretization is given.

**Attention-Based Operators: Data-Driven Compression.** A broad family of operator learners constructs compression/reconstruction from data, i.e., $U(H, X)$ and $V(H, X)$ are feature-dependent:

$$Z = V(H, X)^\top H \in \mathbb{R}^{M \times d}, \quad H' = U(H, X) \widetilde{Z}, \quad (6)$$

where $U, V \in \mathbb{R}^{N \times M}$ are learned compression and reconstruction maps. Linear attention variants, such as Galerkin

Transformer (Cao, 2021) and OFormer (Li et al., 2022a), can be viewed as choosing $U = Q = X W_Q$, $V = K = X W_K$ (with implicit $M = d$) and simplifying the latent mixing $G$ to identity or channel-wise scaling. Structured approaches like FactFormer (Li et al., 2023b) constrain $U, V$ via axial factorization, projecting features onto decoupled 1D axes to approximate high-dimensional integrals.

Recent latent-token operators such as Transolver (Wu et al., 2024) implement the compression map $V^\top$ via *slicing*: they first construct an $N \times M$ soft assignment (slice) matrix that partitions spatial points into $M$ slices, and then aggregate features within each slice to form latent tokens. The slicing weights are produced by a learnable linear map and can be interpreted as attention scores induced by a set of learnable latent queries (see Appendix B for an explicit mapping to standard attention). Transolver reuses the slice assignment for broadcasting latents back to points, which implicitly ties reconstruction and compression (enforcing a symmetry where $U \propto V$). LinearNO (Hu et al., 2025) keeps the slicing-based design but decouples $U$ and $V$ and restricts the latent mixing $G$ to a simplified form. To avoid slice degeneration, additional mechanisms such as learnable temperatures and Gumbel-Softmax are introduced. Although effective, these pipelines typically require explicitly materializing the $N \times M$ weight and applying extra normalization and aggregation outside standard dot-product attention, which makes them difficult to directly leverage highly-optimized compute kernels and tends to be more fragile under half-precision.

Overall, Eq. (4) highlights the desired design: (i) data-driven $U, V$ for diverse geometries, (ii) a powerful latent mixer $G$ for nonlinear correlations, and (iii) an implementation expressed entirely with standard primitives. LRSA is a minimal instantiation of this design.

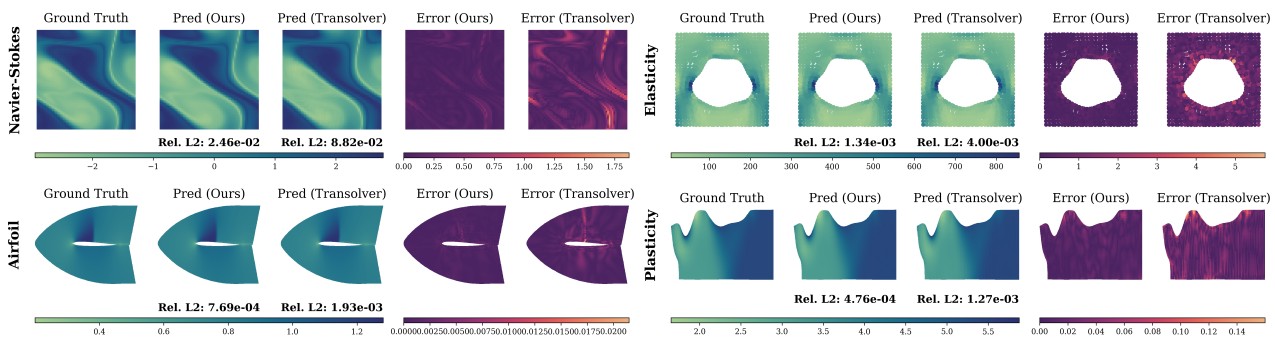

*Figure 3.* **Qualitative performance comparison across diverse discretizations.** From top-left to bottom-right: Navier-Stokes (regular grid), Elasticity (point cloud), Airfoil and Plasticity (structured grid). Error maps are visualized on the same scale for each task. LRSA yields lower relative errors and preserves sharper physical patterns in high-frequency regions compared to Transolver.

### 3.4. Low-Rank Spatial Attention (LRSA)

LRSA instantiates the low-rank template in Eq. (4) with a data-driven basis learned via standard Transformer primitives, while keeping the latent operator $G$ expressive. We use standard scaled dot-product attention (SDPA):

$$\text{Attn}(Q, K, V) = \text{softmax}\left(QK^\top / \sqrt{d_h}\right) V, \quad (7)$$

where the softmax is taken over the key dimension. In practice we employ multi-head attention and head indices are omitted for clarity. All parameters, including the latent queries defined below, are layer-specific.

**Compression** ($V^\top$). LRSA compresses point features into $M$ latent tokens using cross-attention between a set of learnable latent queries $P \in \mathbb{R}^{M \times d}$ and spatial points:

$$Z = \text{Attn}\left(PW_Q^\downarrow, \ HW_K^\downarrow, \ HW_V^\downarrow\right) \in \mathbb{R}^{M \times d}. \quad (8)$$

This produces a compact set of latent tokens as global latent representations, implementing the compression map $V^\top$.

**Latent Mixing** ($G$). We apply a standard Transformer (self-attention and FFN) on the compact tokens:

$$\begin{aligned}
\tilde{Z} &= Z + \text{FFN}_{\text{in}}(\text{Norm}(Z)), \\
\hat{Z} &= \tilde{Z} + \text{SelfAttn}(\text{Norm}(\tilde{Z})), \quad (9) \\
Z' &= \hat{Z} + \text{FFN}_{\text{out}}(\text{Norm}(\hat{Z})).
\end{aligned}$$

where self-attention enables token mixing across latents to capture global correlations, while the FFNs allow the channel mixing within each token. By keeping $M \ll N$, we afford an expressive operator $G$ at low cost.

**Reconstruction** ($U$). We propagate the processed latents back to all points via another cross-attention:

$$\Delta H = \text{Attn}\left(HW_Q^\uparrow, \ Z'W_K^\uparrow, \ Z'W_V^\uparrow\right) \in \mathbb{R}^{N \times d}, \quad (10)$$

followed by a residual update $H' = H + \Delta H$. Importantly, LRSA parameterizes compression and reconstruction with two independent attentions, avoiding tying constraints between writing into and reading from the latent space.

**Complexity and Engineering Advantages.** LRSA replaces point-wise all-to-all interactions with two cross-attentions of width $M$ and a latent self-attention over $M$ tokens. Ignoring constant factors (heads and model width) and noting that the rank $M$ can be kept small regardless of resolution, the computational cost $\mathcal{O}(NM + M^2)$ scales near-linear in $N$. Crucially, LRSA replaces slicing-based tokenization (which explicitly materializes an $N \times M$ assignment matrix and performs additional aggregation/renormalization steps outside the attention primitive) with two standard cross-attention calls and a latent self-attention block. As a result, all aggregation, broadcasting and normalization are handled inside SDPA, enabling direct use of highly-optimized kernels (e.g., FlashAttention-2 (Dao, 2024), Liger-Kernel (Hsu et al., 2025)).

## 4. Experiments

We evaluate LRSA on a diverse suite of benchmarks spanning regular grids, structured meshes, and irregular point clouds. Our experiments aim to verify whether LRSA achieves competitive or superior accuracy across varying geometries compared to strong spectral and attention-based baselines. We also evaluate LRSA's numerical stability and efficiency under mixed-precision, a regime in which existing attention-based operators often fail (Fig. 4). Unless otherwise specified, we report the test relative $L_2$ error ($\|\hat{\mathbf{u}} - \mathbf{u}\|_2 / \|\mathbf{u}\|_2$). To ensure a fair comparison, we align baselines by parameter budget and training configurations. Details on experiments are provided in Appendix D.

### 4.1. Model Accuracy

**Standard Benchmarks.** We evaluate LRSA on six standard PDE benchmarks spanning regular grids (Darcy,

*Table 1.* **Standard Benchmarks.** Comparison of test relative $L_2$ errors on six standard PDE benchmark tasks. Bold indicates the best result, and an underscore indicates the second-best result. "–" indicates the method is not applicable.

| Method | STRUCTURED MESH | | | REGULAR GRID | | POINT CLOUD |
| | AIRFOIL | PIPE | PLASTICITY | NAVIER–STOKES | DARCY | ELASTICITY |
|---|---|---|---|---|---|---|
| FNO (2020b) | – | – | – | 0.1556 | 0.018 | – |
| F-FNO (2023) | 0.0078 | 0.0050 | 0.0047 | 0.2231 | 0.0077 | 0.0263 |
| LSM (2023) | 0.0059 | 0.0050 | 0.0025 | 0.1535 | 0.0065 | 0.0218 |
| HPM (2025) | 0.0047 | 0.0030 | 0.0008 | 0.0734 | 0.0046 | – |
| LNO (2024) | 0.0053 | 0.0031 | 0.0028 | 0.0734 | 0.0063 | 0.0066 |
| Transolver (2024) | 0.0053 | 0.0030 | 0.0013 | 0.0882 | 0.0055 | 0.0065 |
| Transolver++ (2025) | 0.0051 | 0.0027 | 0.0014 | 0.1010 | 0.0056 | 0.0064 |
| LinearNO (2025) | 0.0049 | 0.0024 | 0.0011 | 0.0699 | 0.0050 | 0.0050 |
| Ours | **0.0042** | **0.0023** | **0.0005** | **0.0484** | **0.0044** | **0.0033** |

Navier–Stokes) and structured meshes/point clouds (Airfoil, Plasticity, Pipe, Elasticity). These tasks represent small-to-medium scale regimes with spatial resolutions ranging from $\sim$ 1k to $\sim$ 16k points; full dataset specifications are provided in Appendix C. We compare against two distinct categories of baselines: (i) Spectral-based methods, including FNO variants (Li et al., 2020b; Tran et al., 2023), LSM (Wu et al., 2023), and HPM (Yue et al., 2025), which rely on global frequency or eigen-decompositions; and (ii) Attention-based methods, including Transolver (Wu et al., 2024), LNO (Wang & Wang, 2024), and LinearNO (Hu et al., 2025), which approximate global attention via varying compression strategies. A qualitative comparison across diverse discretizations is illustrated in Figure 3.

Results in Table 1 demonstrate that LRSA achieves the best performance across all six tasks. While the performance gap is narrower on simpler, quasi-static problems (e.g., Pipe and Darcy) where baselines already achieve low errors, LRSA exhibits advantages especially on tasks involving irregular domains or complex dynamics (e.g., Elasticity and Navier–Stokes) compared to the next-best competitors. This improvement suggests that standard, unconstrained attention primitives suffice to achieve high accuracy on complex physical dynamics, without relying on handcrafted heuristics.

**Irregular Benchmarks** We further evaluate LRSA on a suite of irregular domain benchmarks (Chen et al., 2024): Irregular Darcy, Pipe Turbulence, Heat Transfer, and Composite. These tasks feature complex, industrial-style geometries where traditional FFT-based spectral methods are inapplicable. We compare against GraphSAGE (Hamilton et al., 2017), DeepONet (Lu et al., 2019), and two geometry-specialized operators, NORM (Chen et al., 2024) and HPM (Yue et al., 2025). As shown in Table 2, LRSA achieves the best results on most and remains competitive on Irregular Darcy. These findings demonstrate that a purely learnable latent bottleneck can effectively rival specialized

*Table 2.* **Irregular-domain benchmarks.** Relative $L_2$ error on four irregular-domain problems: Irregular Darcy, Pipe Turbulence, Heat Transfer, and Composite. Lower is better.

| Method | Irregular Darcy | Pipe Turbulence | Heat Transfer | Comp. |
|---|---|---|---|---|
| GraphSAGE | 6.73e-2 | 2.36e-1 | – | 2.09e-1 |
| DeepONet | 1.36e-2 | 9.36e-2 | 7.20e-4 | 1.88e-2 |
| NORM | 1.05e-2 | 1.01e-2 | 1.11e-3 | 1.00e-2 |
| HPM | **7.36e-3** | 8.26e-3 | 1.84e-4 | 9.34e-3 |
| Ours | 7.56e-3 | **4.89e-3** | **1.49e-4** | **8.71e-3** |

baselines like NORM and HPM, suggesting that explicit geometric priors (e.g., Laplace eigenfunctions) are not strictly necessary to achieve high accuracy on irregular domains.

**Industrial Cases with Large Geometries.** We assess the model's capability on large-scale aerodynamic simulations using the AirfRANS (Bonnet et al., 2022) and ShapeNet Car (Umetani & Bickel, 2018) datasets. These benchmarks feature high-resolution discretizations ($\sim$ 32k points) and demand precise estimation of derived engineering quantities—such as drag/lift coefficients ($C_D, C_L$), which are often more critical than point-wise errors for design optimization. As reported in Table 3, LRSA matches or exceeds strong baselines across most metrics. On ShapeNet Car, LRSA achieves comparable performance to the best competing methods on both field errors and drag-related metrics. In particular, on AirfRANS, LRSA reduces surface-field error while maintaining near-perfect rank correlation, highlighting its practical reliability on complex, high-resolution airfoil geometries.

### 4.2. Generalization Capabilities

**Zero-Shot Resolution Generalization.** We assess the discretization invariance of LRSA by training on a fixed resolu-

*Table 3.* **Performance comparison on AirFRANS and ShapeNet Car datasets.** In addition to the field-level MSE errors for Volume and Surface quantities, we report the relative errors for lift ($C_L$) and drag ($C_D$) coefficients, along with their Spearman's rank correlations ($\rho_L, \rho_D$). A correlation close to 1.0 indicates that the model correctly preserves design rankings, which is critical for optimization tasks. Lower values ($\downarrow$) are better for error metrics, while higher values ($\uparrow$) are better for correlation coefficients.

| Model | AIRFRANS (BONNET ET AL., 2022) | | | | SHAPENET CAR (UMETANI & BICKEL, 2018) | | | |
|---|---|---|---|---|---|---|---|---|
| | Volume $\downarrow$ | Surface $\downarrow$ | $C_L \downarrow$ | $\rho_L \uparrow$ | Volume $\downarrow$ | Surface $\downarrow$ | $C_D \downarrow$ | $\rho_D \uparrow$ |
| GraphSAGE (2017) | 0.0087 | 0.0184 | 0.1476 | 0.9964 | 0.0461 | 0.1050 | 0.0270 | 0.9695 |
| GraphUNet (2019) | 0.0076 | 0.0144 | 0.1677 | 0.9949 | 0.0471 | 0.1102 | 0.0226 | 0.9725 |
| MeshGraphNet (2020) | 0.0214 | 0.0387 | 0.2252 | 0.9945 | 0.0354 | 0.0781 | 0.0168 | 0.9840 |
| GNO (2020a) | 0.0269 | 0.0405 | 0.2016 | 0.9938 | 0.0383 | 0.0815 | 0.0172 | 0.9834 |
| GEO-FNO (2022b) | 0.0361 | 0.0301 | 0.6161 | 0.9257 | 0.1670 | 0.2378 | 0.0664 | 0.8280 |
| LNO (2024) | 0.0214 | 0.0268 | 0.1480 | 0.9744 | 0.0269 | 0.0870 | 0.0174 | 0.9781 |
| Transolver (2024) | 0.0023 | 0.0085 | 0.1230 | 0.9978 | 0.0221 | 0.0785 | 0.0117 | 0.9914 |
| LinearNO (2025) | 0.0011 | 0.0077 | 0.0491 | 0.9992 | 0.0194 | 0.0754 | **0.0106** | **0.9925** |
| **Ours** | **0.0011** | **0.0010** | **0.0396** | **0.9997** | **0.0166** | **0.0719** | **0.0106** | **0.9925** |

tion ($221 \times 51$) and evaluating on unseen grids with varying densities. As illustrated in Table 4, while all models naturally degrade on unseen resolutions, LRSA demonstrates superior stability, confirming its capability to approximate the underlying continuous operator rather than overfitting to the specific training grid structure.

*Table 4.* Relative $L_2$ error ($\times 10^{-2}$) across evaluation meshes of different spatial resolutions. All models are trained on the $221 \times 51$ mesh and evaluated on the resolutions listed in the columns. Lower is better.

| Model | $221 \times 51$ | $111 \times 51$ | $221 \times 26$ | $111 \times 26$ |
|---|---|---|---|---|
| Transolver | 0.524 | 7.850 | 1.260 | 7.680 |
| HPM | 0.438 | 0.537 | 1.690 | 1.740 |
| Ours | **0.423** | **0.468** | **0.644** | **0.673** |

**Limited Training Data.** We assess sample efficiency by training on subsets of the Darcy and Navier–Stokes problems, as presented in Table 5. On Navier–Stokes with 200 training samples, HPM achieves lower error than LRSA. This outcome is consistent with the use of stronger heuristic geometric and spectral priors in HPM, which may be advantageous in the extreme low-data regime. As the training set increases, LRSA improves more rapidly and achieves the best performance among the compared methods, indicating favorable scaling with available supervision.

### 4.3. Ablation Study

**Component Analysis.** We ablate two design choices in LRSA to isolate their contributions while keeping the down-/up- cross-attention structure unchanged. (i) *w/o Intra Attn*: we replace the latent processing blocks with an MLP, removing the self-attention. (ii) *Symmetric*: we tie the

*Table 5.* Relative $L_2$ errors ($\times 10^{-2}$) on Darcy and Navier–Stokes under limited training data. The second column reports the number of training samples used for each problem. Lower is better.

| Prob. | Training Number | Transolver | LNO | HPM | Ours |
|---|---|---|---|---|---|
| Darcy | 200 | 1.75 | 5.62 | 1.10 | **0.97** |
| | 600 | 0.69 | 4.40 | 0.60 | **0.50** |
| | 1000 | 0.52 | 0.63 | 0.44 | **0.43** |
| NS | 200 | 37.6 | 40.0 | **18.1** | 26.2 |
| | 600 | 31.4 | 25.0 | 11.7 | **7.50** |
| | 1000 | 9.60 | 7.34 | 7.44 | **4.84** |

down-projection keys and up-projection queries, enforcing a shared subspace basis (analogous to Transolver). The results are summarized in Figure 5 (Bottom). Across benchmarks, removing the latent blocks consistently increases errors, particularly in time-dependent problems, suggesting that explicit mixing in the latent space is necessary for capturing complex dynamic correlations. Furthermore, enforcing symmetry results in comparable or even higher errors than removing latent self-attention. This suggests that the optimal subspace for compressing input features differs inherently from the basis required for reconstruction, validating our choice to decouple the encoding and decoding transformations.

**Latent Rank.** We study the effect of the latent rank $M$, i.e., the bottleneck width in LRSA (number of latent queries) and the number of slices in Transolver (Figure 5, Top). Empirically, the error typically saturates once $M$ exceeds a moderate value on several benchmarks, suggesting that the layer-wise global coupling can be captured by a compact set of latent modes (i.e., low effective rank) in practice.

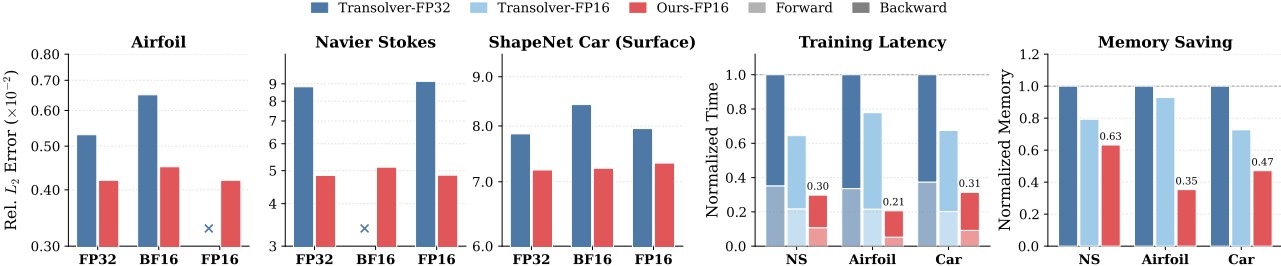

*Figure 4.* **Training stability and efficiency.** Left: relative $L_2$ error under FP32/BF16/FP16; $\times$ denotes divergence. Right: per-step training latency (forward+backward, normalized to Transolver-FP32) and peak-memory (ratio relative to Transolver-FP32; smaller is better) on three representative tasks. Memory Saving is calculated as the ratio of peak training memory consumption of the evaluated model to that of the baseline model (Transolver-FP32). A higher factor indicates superior memory efficiency.

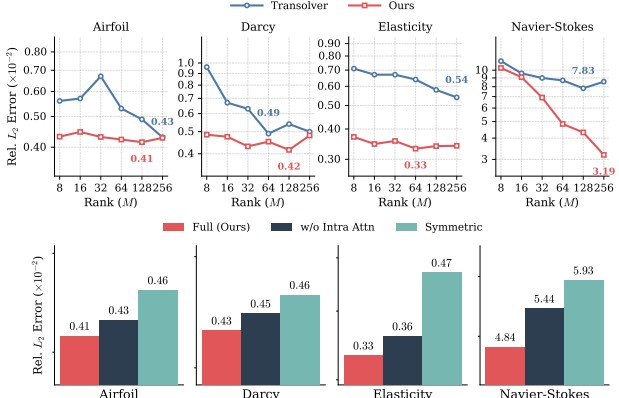

*Figure 5.* **Rank and component ablations.** Top: sensitivity to latent size $M$. Bottom: component variants of LRSA (Full, w/o latent self-attention, and enforcing symmetric down/up projections).

This provides direct experimental support for the low-rank bottleneck assumption underlying LRSA, without requiring explicit kernel materialization. On complex dynamics (Navier–Stokes), LRSA scales more effectively with $M$: increasing $M$ yields a steep error reduction, whereas Transolver improves much more slowly with additional slices. Conversely, on simpler static fields, LRSA reaches near-optimal performance with few latents ($M \approx 32$); further increasing $M$ shows diminishing returns and mild overfitting, while Transolver typically needs larger $M$ to match the same accuracy. In practice, $M = 64$ is a robust default; we recommend checking $M = 32$ for simpler static problems and $M = 128$ when validation error suggests under-capacity.

**Training Stability & Efficiency.** Compatibility with standard hardware-optimized kernels gives LRSA a decisive advantage in both numerical stability and computational efficiency. As shown in Figure 4 (Left), the baseline Transolver is highly sensitive to numerical precision: it suffers from significant error increases in BFloat16 (BF16) and frequently

diverges in Float16 (FP16). We attribute this instability to its explicit slice normalization, which requires a high dynamic range to prevent denominator underflow. In contrast, LRSA leverages standard Transformer primitives supported by robust fused attention kernels, maintaining near-consistent accuracy across FP32, BF16, and FP16. This stability supports mixed-precision training (Figure 4, Right). In our experiments, Transolver is more reliable in FP32, while LRSA also works well in FP16. On ShapeNet Car ($N \approx 32$k), LRSA-FP16 achieves $3.2\times$ lower training latency and $2.1\times$ lower peak memory than Transolver-FP32. Tables 6 and 7 show that, under matched FP16 settings, LRSA has the lowest cost among similarly accurate attention-based baselines we evaluate, and that its cost remains near-linear beyond 32k points, making higher-resolution training practical on a single modern GPU.

*Table 6.* **Absolute FP16 inference efficiency.** Comparison with recent attention-based baselines using official implementations on a single RTX 3090 GPU with $M = 64$. Lower is better for FLOPs, latency, and memory.

| Method | Params | FLOPs (sample) | Latency (ms) | Mem (MB) |
|---|---|---|---|---|
| *Navier Stokes $N = 64 \times 64$ (batch size 20)* | | | | |
| LRSA | 12.35M | **36.5B** | **34.8** | **352** |
| LinearNO | 10.19M | 91.8B | 83.2 | 678 |
| Transolver++ | 14.36M | 120.3B | 66.7 | 487 |
| *ShapeNet Car $N = 32$k (batch size 2)* | | | | |
| LRSA | 9.14M | **206B** | **36.7** | **218** |
| LinearNO | 6.00M | 449B | 282.8 | 661 |
| Transolver++ | 5.46M | 378B | 42.4 | 439 |

## 5. Conclusion

We presented a unified low-rank view of global mixing in neural operators, connecting spectral/basis approaches and latent-token attention models under a common template. Building on this perspective, we introduced Low-Rank Spatial Attention (LRSA), a minimal block that routes

*Table 7.* **Scaling beyond 32k points.** Results for a 6-layer, width-128 LRSA model on a single RTX 3090 GPU with batch size 1. Lower is better for latency and memory.

| Points | Inference ($N$) | Training (ms) | Train Mem (MB) |
|--------|-----------------|---------------|----------------|
| 128k   | 71.4            | 439.9         | 5,010          |
| 256k   | 138.1           | 890.9         | 9,980          |
| 512k   | 274.8           | 1,797.1       | 19,920         |

global interactions through a compact set of learnable latents using only standard Transformer primitives. This design directly benefits from hardware-optimized computational kernels, enabling robust mixed-precision training without custom routing or normalization components. Across regular grids, structured meshes, and irregular point clouds, LRSA achieves strong accuracy and improves over competitive neural-operator baselines. One limitation is that a fixed latent bottleneck can under-resolve sharply localized structures, such as boundary layers or thin interior interfaces, where errors tend to concentrate near steep gradients. Another remaining challenge is to better understand how to allocate model capacity between the latent processing and point-wise updates as resolution and task complexity increase. Exploring principled capacity allocation strategies, potentially guided by profiling and large-scale multi-physics pretraining, is a promising direction for making operator learning even more practical.

## Impact Statement

This work aims to advance neural operator learning for scientific simulation. There are no specific societal consequences that require individual highlighting here.

## Acknowledgements

This work was supported by the National Natural Science Foundation of China (62025207). Tao Du acknowledges the support from Tsinghua University and the Shanghai Qi Zhi Institute Innovation Program. We also thank the anonymous reviewers for their valuable feedback and suggestions.

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

# A. LRSA as a Low-Rank Integral Operator

We provide an intuitive operator-level interpretation of LRSA. The uniform-sampling assumption introduced below is used only to interpret the down-projection as a quadrature approximation of a spatial integral. The latent processing and up-projection stages remain well-defined and resolution-invariant regardless of the point distribution. Let $\Omega \subset \mathbb{R}^{d_{\text{phys}}}$ be a bounded domain, and let the (lifted) feature field be a function $h : \Omega \to \mathbb{R}^d$. We assume the input points $\{x_i\}_{i=1}^N$ are uniformly sampled in $\Omega$, so that a simple quadrature rule applies:

$$\int_\Omega \phi(y)\, dy \;\approx\; \frac{|\Omega|}{N} \sum_{i=1}^N \phi(x_i), \qquad \text{for sufficiently regular } \phi. \tag{11}$$

Below we interpret each LRSA stage as a continuous operator, with the discrete implementation being its numerical (quadrature) approximation.

**Continuous Softmax Attention**   For a fixed query vector $q \in \mathbb{R}^{d_h}$ and key/value fields $k(\cdot), v(\cdot)$, define the continuous softmax attention operator:

$$\text{Attn}_\Omega(q; k, v) \;=\; \int_\Omega a_q(y)\, v(y)\, dy, \qquad a_q(y) \;=\; \frac{\exp(\langle q, k(y)\rangle)}{\int_\Omega \exp(\langle q, k(z)\rangle)\, dz}. \tag{12}$$

This defines a well-posed mapping from an input function $v(\cdot)$ to an output vector, and is independent of any discretization.

**Step 1 (Down): Compression as a Continuous Projection**   In LRSA, the down-projection uses $M$ learnable latent queries $\{p_m\}_{m=1}^M$ and cross-attention from latents to points. Absorbing linear projections into the definitions $k_\downarrow(y) = h(y)W_K^\downarrow$ and $v_\downarrow(y) = h(y)W_V^\downarrow$, the continuous counterpart of the $m$-th latent token is:

$$L_m \;=\; \text{Attn}_\Omega(p_m; k_\downarrow, v_\downarrow) \;=\; \int_\Omega \alpha_m(y; h)\, v_\downarrow(y)\, dy, \tag{13}$$

where $\alpha_m(\cdot; h)$ is the normalized exponential weight in Eq. (12). The discrete down-projection in implementation is exactly Eq. (13) with the integral replaced by the quadrature in Eq. (11). Hence, under approximately uniform sampling, the discrete LRSA down-stage approximates the same continuous mapping.

**Step 2 (Latent Processing): Resolution-Invariant Mixing in a Fixed Latent Space**   The latent processing stage applies a standard Transformer (self-attention + FFN) on the $M$ tokens:

$$L' = \mathcal{T}_\theta(L), \qquad \mathcal{T}_\theta : \mathbb{R}^{M \times d} \to \mathbb{R}^{M \times d}. \tag{14}$$

Crucially, Eq. (14) depends only on $M$ (fixed) and does not depend on the spatial resolution $N$. Thus it defines a resolution-invariant nonlinear map on a compact set of global coefficients.

**Step 3 (Up): Reconstruction as a Finite Expansion over $M$ Latent Modes**   The up-projection broadcasts latent information back to any query location $x \in \Omega$ via cross-attention from points to latents. Let $q_\uparrow(x) = h(x)W_Q^\uparrow$ and define latent keys/values $k_\uparrow(m) = L_m' W_K^\uparrow$, $v_\uparrow(m) = L_m' W_V^\uparrow$. Then the continuous update at location $x$ is:

$$\Delta h(x) \;=\; \sum_{m=1}^M \beta_m(x; L')\, v_\uparrow(m), \qquad \beta_m(x; L') = \frac{\exp(\langle q_\uparrow(x), k_\uparrow(m)\rangle)}{\sum_{r=1}^M \exp(\langle q_\uparrow(x), k_\uparrow(r)\rangle)}. \tag{15}$$

This is a finite expansion over $M$ latent modes, valid for any continuous coordinate $x$ (no discretization is required for defining Eq. (15)).

**Overall Operator and Low-Rank Structure**   Composing Eq. (13), (14), and (15), LRSA defines an operator $\mathcal{G}_\theta$ acting on the function $h(\cdot)$:

$$(\mathcal{G}_\theta h)(x) \;=\; h(x) + \Delta h(x), \qquad \Delta h(x) = \sum_{m=1}^M \beta_m(x; \mathcal{T}_\theta(L))\, v_\uparrow(m), \quad L_m = \int_\Omega \alpha_m(y; h)\, v_\downarrow(y)\, dy. \tag{16}$$

To make the low-rank nature explicit, consider the induced interaction between an output location $x$ and input location $y$. LRSA routes all global coupling through only $M$ latent channels, yielding an effective (input-dependent) kernel of the form:

$$\kappa_\theta(x, y; h) \approx \sum_{m=1}^{M} \sum_{k=1}^{M} \beta_m(x; h)\, G_{mk}(h)\, \alpha_k(y; h), \tag{17}$$

where $G_{mk}(h)$ summarizes the latent-space mixing induced by $\mathcal{T}_\theta$. Equation (17) is a separable expansion in $(x, y)$ with at most $M$ latent modes, i.e., a finite-rank (low-rank) approximation of a dense global interaction map.

In summary, under uniform sampling, the down-stage of the discrete LRSA block is a numerical (quadrature) approximation of the continuous operator in Eq. (16), while the latent processing and up-stage remain resolution-invariant by construction. Its global coupling is mediated by $M \ll N$ latent modes, leading to a low-rank induced kernel and near-linear complexity in the number of spatial samples.

## B. A Unified Low-Rank View for Neural Operators

### B.1. Spectral / Basis-Based Operators

We show that spectral and basis-expansion neural operators can be interpreted as a special case of the low-rank template in Eq. (4). The key observation is that truncating to $M \ll N$ basis functions induces a rank-$M$ (or low effective-rank) approximation of a dense global interaction map.

**From Integral Operators to Truncated Basis Expansions.** Many PDE solution operators can be written as an integral operator

$$(\mathcal{K}f)(x) = \int_\Omega \kappa(x, y)\, f(y)\, dy, \tag{18}$$

where $\kappa$ is a (typically smooth) kernel. A classical approximation expands $\kappa$ (or the solution field) in a basis $\{\phi_m\}_{m=1}^\infty$ and truncates to the first $M$ modes:

$$\kappa(x, y) \approx \sum_{m=1}^{M} \sum_{n=1}^{M} \phi_m(x)\, G_{mn}\, \phi_n(y). \tag{19}$$

This yields the separable form that directly corresponds to a low-rank factorization.

**Discrete Low-Rank Factorization.** Given $N$ sample points $\{x_i\}_{i=1}^N$, define the basis evaluation matrix $\Phi \in \mathbb{R}^{N \times M}$ with $\Phi_{i,m} = \phi_m(x_i)$. Discretizing Eq. (19) gives a global mixing layer

$$H' = \Phi\, G\, \Phi^\top H, \tag{20}$$

which matches Eq. (4) with

$$U = \Phi, \qquad V = \Phi, \qquad \text{and} \qquad G \in \mathbb{R}^{M \times M}. \tag{21}$$

Therefore, spectral/basis-based operators realize global coupling by (i) projecting to a compact basis ($\Phi^\top$), (ii) mixing in the basis space ($G$), and (iii) reconstructing back to the spatial domain ($\Phi$). When $M \ll N$, the induced interaction map is (at most) rank-$M$.

**Instantiation in Representative Operators. FNO.** On regular grids, the basis $\Phi$ corresponds to the discrete Fourier transform (DFT) matrix. Frequency truncation in FNO keeps only $M$ low-frequency modes, making Eq. (20) efficient. The latent operator $G$ is implemented by learnable channel mixing on the retained Fourier coefficients (often parameterized per frequency).

**Geometry-induced spectral methods (e.g., NORM/HPM).** On irregular domains, $\Phi$ can be constructed from the first $M$ eigenfunctions of the Laplace–Beltrami operator evaluated on the mesh/point set. This again yields a low-rank global mixing of the form Eq. (20), where the projection and reconstruction are determined by a geometry-dependent prior basis.

**Relation to LRSA.** Eq. (20) highlights a key difference between basis operators and LRSA. Basis operators use an *analytic* (Fourier) or *geometry-induced* (eigenfunction) $\Phi$ that is fixed once the discretization/geometry is chosen. In contrast, LRSA learns a data-driven analogue of such a compact basis via latent queries and standard attention, without requiring FFT structure, mesh connectivity, or precomputed eigenbases.

## B.2. Attention-Based Operators

To contextualize our proposed Low-Rank Spatial Attention (LRSA) within the literature, we explicitly map the *Physics-Attention* mechanism from Transolver (Wu et al., 2024) into our generalized low-rank template $K \approx UGV^\top$. We demonstrate that Transolver can be viewed as a restricted instance of LRSA characterized by (i) specific parameter tying and (ii) a symmetric constraint on the factorization.

**Slicing as Latent-Query Attention.** In Transolver, the compression of spatial points $X \in \mathbb{R}^{N \times d}$ into $M$ latent tokens (slices) $L \in \mathbb{R}^{M \times d}$ is defined by learnable projection weights $W_{\text{slice}} \in \mathbb{R}^{M \times d}$. The assignment scores $A \in \mathbb{R}^{N \times M}$ are computed as:

$$A = \text{Softmax}(XW_{\text{slice}}^\top), \tag{22}$$

where the softmax is typically applied across the slice dimension $M$. Let us interpret the rows of $W_{\text{slice}}$ as a set of *latent queries* $P \in \mathbb{R}^{M \times d}$. If we formulate a standard cross-attention where $P$ queries the points $X$, the attention score matrix $S \in \mathbb{R}^{M \times N}$ would be:

$$S = \text{Softmax}\left(\frac{PX^\top}{\sqrt{d}}\right). \tag{23}$$

Note that Transolver's scores $A$ are simply $(PX^\top)^\top$ with a different normalization axis. Consequently, the "Physics-aware Token Generation" in Transolver:

$$z_j = \frac{\sum_{i=1}^{N} w_{i,j} x_i}{\sum_{i=1}^{N} w_{i,j}} \tag{24}$$

is mathematically equivalent to an attention operation where query-projection, key-projection, and value-projection are all identity mappings of $W_{\text{slice}}$ and $X$, and the softmax is normalized per query. Thus, Transolver's slicing is a manual realization of Step 1 in LRSA (Down-Projection), but with fewer learnable degrees of freedom.

**Symmetry vs. Decoupling.** The structural template $K \approx UGV^\top$ reveals the core limitation of Transolver's reconstruction (Up-Projection). Transolver broadcasts latent information back to spatial points using the *same* assignment matrix $A$ calculated during the down-projection:

$$\Delta X = A\hat{L}, \tag{25}$$

where $\hat{L}$ are the processed latent tokens. In our low-rank framework, this corresponds to enforcing a *symmetric constraint* on the factorization, such that:

$$U = V \propto A. \tag{26}$$

This implies that the affinity of a spatial point $x_i$ to a latent token $l_j$ must be identical for both *writing* information to and *reading* information from the latent space.

In contrast, LRSA decouples these operators using two independent cross-attentions. Step 1 learns the compression basis $V^\top$, while Step 3 (Up-Projection) learns a separate reconstruction basis $U$:

$$\Delta H = \text{Attn}(Q = HW_Q^{\text{up}}, K = L'W_K^{\text{up}}, V = L'W_V^{\text{up}}). \tag{27}$$

This decoupling allows the model to interpret latent tokens differently at each stage—for example, a token might capture high-frequency features in the down-stage but distribute them selectively based on global context in the up-stage.

**Engineering Consequences.** Beyond the mathematical formulation, Transolver's manual implementation of $A = \text{Softmax}(XW^\top)$ and the subsequent normalization and up projection requires materializing the dense $N \times M$ weight matrix and necessitating FP32 precision to maintain numerical stability. By reformulating these interactions as strictly standard attention blocks, LRSA (i) inherits the numerical stability improvements and (ii) achieves higher throughput via out-of-the-box compatibility with hardware-optimized kernels like FlashAttention, which are not designed for the specific weight-tying required by Transolver.

**Relation to LinearNO: The Collapse of Latent Mixing.** LinearNO (Hu et al., 2025) reformulates the token-based operator as a kernelized linear attention mechanism. In our low-rank framework $K \approx UGV^\top$, LinearNO can be interpreted as a factorization where the latent processing operator $G$ is restricted to be linear or identity, effectively removing the non-linear mixing capability within the latent space.

Specifically, LinearNO computes the output as:

$$\text{LinearNO}(X) = \underbrace{\phi(XW_Q)}_{U} \cdot \left( \underbrace{\psi(XW_K)^\top}_{V^\top} \cdot (XW_V) \right), \tag{28}$$

where $\phi(\cdot)$ and $\psi(\cdot)$ are kernel functions (e.g., softmax normalized along specific axes).

- **Decoupling (Step 1 & 3):** Unlike Transolver, LinearNO employs separate projections for $\phi$ and $\psi$, meaning it successfully decouples the down-projection basis $V^\top$ from the up-projection basis $U$. This aligns with LRSA's design and avoids the symmetric constraint.

- **Latent Collapse (Step 2):** The critical distinction lies in the intermediate term. In LRSA, the latent features $L = V^\top X$ undergo a deep, non-linear transformation via Self-Attention and FFNs ($G_{\text{nonlinear}}$) before reconstruction. LinearNO, aiming for maximum efficiency, performs the reconstruction directly after aggregation. This implies $G \approx I$ (identity) or a simple linear map.

Our ablation study demonstrates that while this linearization is sufficient for simple static problems, retaining the non-linear Latent Transformer ($G$) is essential for capturing complex, time-dependent dynamics (e.g., Navier-Stokes), justifying our choice to keep the bottleneck expressive.

## C. Dataset and Evaluation Metrics

### C.1. Standard Benchmarks

Consider the boundary-value problem given by:

$$\begin{aligned} Lu = f, & \quad \text{in } \Omega, \\ u = g, & \quad \text{on } \partial\Omega. \end{aligned} \tag{29}$$

Our experiments cover variables including: (1) coefficients in $L$, describing material properties; (2) external forcing $f$; (3) domain geometry $\Omega$; and (4) boundary shape $\partial\Omega$. We also consider time-dependent problems where current states determine future evolution. A summary of the datasets is provided in Table 8.

*Table 8.* **Overview of Standard Benchmarks.** Details on geometry type, spatial dimension, discretization size, dataset splits, and input features.

| Test Case | Geometry | #Dim | Mesh Size | Dataset Split Train | Dataset Split Test | Input Type |
|---|---|---|---|---|---|---|
| Elasticity (Li et al., 2020b) | Point Cloud | 2 | 972 | 1000 | 200 | Domain Shape |
| Darcy (Li et al., 2020b) | Regular Grid | 2 | $85 \times 85$ | 1000 | 200 | Coefficients |
| Navier Stokes (Li et al., 2020b) | Regular Grid | 2+1 | $64 \times 64$ | 1000 | 200 | Previous States |
| Plasticity (Li et al., 2022b) | Structured Mesh | 2+1 | $101 \times 31$ | 900 | 80 | External Force |
| Airfoil (Li et al., 2022b) | Structured Mesh | 2 | $221 \times 51$ | 1000 | 200 | Boundary Shape |
| Pipe (Li et al., 2022b) | Structured Mesh | 2 | $129 \times 129$ | 1000 | 200 | Domain Shape |

**Elasticity.** This benchmark estimates the inner stress of an elastic material based on its structure, discretized into 972 points (Li et al., 2020b). The input is a tensor of shape $972 \times 2$ containing the 2D coordinates of the discretized points. The output is the stress value at each point ($972 \times 1$).

**Darcy.** This measures fluid flow through a porous medium:

$$\begin{aligned} -\nabla \cdot (a(x)\nabla u(x)) = f(x), & \quad x \in (0,1)^2, \\ u(x) = 0, & \quad x \in \partial(0,1)^2, \end{aligned} \tag{30}$$

where $a(x)$ is the diffusion coefficient (input) and $u(x)$ is the solution (output). The original resolution is $421 \times 421$; we perform $5\times$ downsampling to $85 \times 85$.

**Navier–Stokes.** This models incompressible viscous flow:

$$
\begin{aligned}
\partial_t w(x,t) + u(x,t) \cdot \nabla w(x,t) &= \nu \nabla w(x,t) + f(x), & x &\in (0,1)^2, t \in (0,T], \\
\nabla \cdot u(x,t) &= 0, & x &\in (0,1)^2, t \in (0,T], \\
w(x,0) &= w_0(x), & x &\in (0,1)^2,
\end{aligned}
\tag{31}
$$

where $u$ is velocity and $w = \nabla \times u$ is vorticity. We use $\nu = 10^{-5}$. The dataset contains vorticity fields over 20 time steps. The model takes 10 previous steps to predict the next step.

**Plasticity.** The task is to predict the deformation of a plastic material under impact. Since the initial state is static, the model takes time $t$ and top-boundary conditions as input to predict the deformation field.

**Airfoil.** The Euler equation models transonic flow over an airfoil. The input consists of the mesh grid coordinates representing the airfoil geometry, and the output is the Mach number field defined on the same structured mesh.

**Pipe.** The goal is to predict the $x$-component of fluid velocity in variable-shaped pipes. The input is the structured mesh coordinates of the deformed pipe, and the output is the velocity field.

### C.2. Irregular Domains

These benchmarks involve complex geometries where standard regular grids are inapplicable (Table 9). **Irregular Darcy.**

*Table 9.* **Overview of Irregular Domain Benchmarks.**

| Test Case | Geometry | # Dim | Num. Nodes ($N$) | Dataset Split Train | Dataset Split Test | Input Type |
|-----------|----------|-------|------------------|-------|------|------------|
| Irregular Darcy Chen et al. (2024) | Point Cloud | 2 | 2290 | 1000 | 200 | Coefficients |
| Pipe Turbulence Chen et al. (2024) | Point Cloud | 2 | 2673 | 300 | 100 | Previous State |
| Heat Transfer Chen et al. (2024) | Point Cloud | 3 | 7199 | 100 | 100 | Boundary Shape |
| Composite Chen et al. (2024) | Point Cloud | 3 | 8232 | 400 | 100 | Temperature |

This problem involves solving the Darcy Flow equation within an irregular domain. The function input is $a(x)$, representing the diffusion coefficient field, and the output $u(x)$ represents the pressure field. The domain is represented by a triangular mesh with 2290 nodes. The neural operators are trained on 1000 trajectories and tested on an extra 200 trajectories.

**Pipe Turbulence.** Pipe Turbulence system is modeled by the Navier-Stokes equation, with an irregular pipe-shaped computational domain represented as 2673 triangular mesh nodes. This task requires the neural operator to predict the next frame's velocity field based on the previous one. Same as Chen et al. (2024), we utilize 300 trajectories for training and then test the models on 100 samples.

**Heat Transfer.** This problem is about heat transfer events triggered by temperature variances at the boundary. Guided by the Heat equation, the system evolves over time. The neural operator strives to predict 3-dimensional temperature fields after 3 seconds given the initial boundary temperature status. The output domain is represented by triangulated meshes of 7199 nodes. The neural operators are trained on 100 data sets and evaluated on another 100 data.

**Composite.** This problem involves predicting deformation fields under high-temperature stimulation, a crucial factor in composite manufacturing. The trained operator is anticipated to forecast the deformation field based on the input temperature field. The structure studied in this paper is an air-intake component of a jet composed of 8232 nodes, as referenced in (Chen et al., 2024). The training involved 400 data, and the test examined 100 data.

### C.3. Industrial Cases with Large Geometries

**ShapeNet Car** This benchmark focuses on the drag coefficient estimation for the driving car, which is essential for car design. Overall, 889 samples with different car shapes are generated to simulate the 72 km/h speed driving situation (Umetani & Bickel, 2018), where the car shapes are from the "car" category of ShapeNet (Chang et al., 2015). Concretely,

| Test Case | Geometry | #Dim | Mesh Size | Dataset Split | | Input Type |
|---|---|---|---|---|---|---|
| | | | | Train | Test | |
| ShapeNet Car (Umetani & Bickel, 2018) | Unstructured Mesh | 3 | 32186 | 789 | 100 | Structure |
| AirfRANS (Bonnet et al., 2022) | Unstructured Mesh | 2 | 32000 | 800 | 200 | Structure |

*Table 10.* Datasets and their split used in our experiments.

they discretize the whole space into unstructured mesh with 32,186 mesh points and record the air around the car and the pressure over the surface. The input mesh of each sample is also preprocessed into the combination of mesh point position, signed distance function and normal vector. The model is trained to predict the velocity and pressure value for each point. Afterward, we can calculate the drag coefficient based on these estimated physics fields.

**AirfRANS** This dataset contains the high-fidelity simulation data for Reynolds-Averaged Navier–Stokes equations (Bonnet et al., 2022), which is also used to assist airfoil design. Different from Airfoil (Li et al., 2023a), this benchmark involves more diverse airfoil shapes under finer discretized meshes. Specifically, it adopts airfoils in the 4 and 5 digits series of the National Advisory Committee for Aeronautics, which have been widely used historically. Each case is discretized into 32,000 mesh points. By changing the airfoil shape, Reynolds number, and angle of attack, AirfRANS provides 1000 samples, where 800 samples are used for training and 200 for the test set. Air velocity, pressure and viscosity are recorded for surrounding space and pressure is recorded for the surface. Note that both drag and lift coefficients can be calculated based on these physics quantities. However, as their original paper stated, air velocity is hard to estimate for airplanes, making all the deep models fail in drag coefficient estimation (Bonnet et al., 2022). Thus, in the main text, we focus on the lift coefficient estimation and the pressure quantity on the volume and surface, which is essential to the take-off and landing stages of airplanes.

### C.4. Metrics

Since our experiment consists of standard benchmarks, Irregular Domains and Industrial Cases with Large Geometries, we also include several design-oriented metrics in addition to the mean squared error for physics fields.

**Mean Squared Error for Physics Fields** Given a dataset of $N$ test samples, where the ground-truth physics field $\mathbf{u}^{(i)}(\mathbf{x}) \in \mathbb{R}^C$ and the model-predicted field $\widehat{\mathbf{u}}^{(i)}(\mathbf{x})$ correspond to the $i$-th sample and are defined over the spatial domain $\Omega$, the mean squared error (MSE) is defined as

$$\text{MSE} = \frac{1}{N} \sum_{i=1}^{N} \left\| \mathbf{u}^{(i)} - \widehat{\mathbf{u}}^{(i)} \right\|_2^2, \tag{32}$$

where $\| \cdot \|_2$ denotes the $\ell_2$ norm over the spatial domain and physical channels.

**Relative L2 for Physics Fields** Given a dataset of $N$ test samples, where the ground-truth physics field $\mathbf{u}^{(i)}(\mathbf{x}) \in \mathbb{R}^C$ and the model-predicted field $\widehat{\mathbf{u}}^{(i)}(\mathbf{x})$ correspond to the $i$-th sample and are defined over the spatial domain $\Omega$, the relative $\mathcal{L}_2$ error is computed as the average of per-sample relative errors:

$$\text{Relative L2} = \frac{1}{N} \sum_{i=1}^{N} \frac{\left\| \mathbf{u}^{(i)} - \widehat{\mathbf{u}}^{(i)} \right\|_2}{\left\| \mathbf{u}^{(i)} \right\|_2}, \tag{33}$$

where $\| \cdot \|_2$ denotes the $\ell_2$ norm over the spatial domain and physical channels. For time-dependent problems, the reported metric is computed based on the rollout predictions over the entire temporal horizon.

$L_{\mathbf{g}}$ **for Physics Fields** Given a dataset of $N$ test samples, where the ground-truth physics field $\mathbf{u}^{(i)}(\mathbf{x}) \in \mathbb{R}^C$ and the model-predicted field $\widehat{\mathbf{u}}^{(i)}(\mathbf{x})$ correspond to the $i$-th sample and are defined over the spatial domain $\Omega$, the $\mathcal{L}_{\mathbf{g}}$ error is computed as the average of per-sample relative errors:

$$L_{\mathbf{g}} = \frac{1}{N} \sum_{i=1}^{N} \frac{\left\| \nabla \mathbf{u}^{(i)} - \nabla \widehat{\mathbf{u}}^{(i)} \right\|_2}{\left\| \nabla \mathbf{u}^{(i)} \right\|_2}, \tag{34}$$

where $\| \cdot \|_2$ denotes the $\ell_2$ norm over the spatial domain and physical channels.

**Relative L2 for Drag and Lift Coefficients**    For Shape-Net Car and AirfRANS, we also calculated the drag and lift coefficients based on the estimated physics fields. For unit density fluid, the coefficient (drag or lift) is defined as follows:

$$C = \frac{2}{v^2 A} \left( \int_{\partial\Omega} p(\boldsymbol{\xi}) \left( \widehat{n}(\boldsymbol{\xi}) \cdot \widehat{i}(\boldsymbol{\xi}) \right) \mathrm{d}\boldsymbol{\xi} + \int_{\partial\Omega} \tau(\boldsymbol{\xi}) \cdot \widehat{i}(\boldsymbol{\xi}) \mathrm{d}\boldsymbol{\xi} \right), \tag{35}$$

where $v$ is the speed of the inlet flow, $A$ is the reference area, $\partial\Omega$ is the object surface, $p$ denotes the pressure function, $\widehat{n}$ means the outward unit normal vector of the surface, $\widehat{i}$ is the direction of the inlet flow and $\tau$ denotes wall shear stress on the surface. $\tau$ can be calculated from the air velocity near the surface (McCormick, 1994), which is usually much smaller than the pressure item. Specifically, for the drag coefficient of Shape-Net Car, $\widehat{i}$ is set as $(-1, 0, 0)$ and $A$ is the area of the smallest rectangle enclosing the front of cars. As for the lift coefficient of AirfRANS, $\widehat{i}$ is set as $(0, 0, -1)$. The relative L2 is defined between the ground truth coefficient and the coefficient calculated from the predicted velocity and pressure field.

**Spearman's Rank Correlations for Drag and Lift Coefficients**    Given $K$ samples in the test set with the ground truth coefficients $C = \{C^1, \cdots, C^K\}$ (drag or lift) and the model predicted coefficients $\widehat{C} = \{\widehat{C}^1, \cdots, \widehat{C}^K\}$, the Spearman correlation coefficient is defined as the Pearson correlation coefficient between the rank variables, that is:

$$\rho = \frac{\mathrm{cov}\left( R(C) R(\widehat{C}) \right)}{\sigma_{R(C)} \sigma_{R(\widehat{C})}}, \tag{36}$$

where $R$ is the ranking function, $\mathrm{cov}$ denotes the covariance and $\sigma$ represents the standard deviation of the rank variables. Thus, this metric is highly correlated to the model guide for design optimization. A higher correlation value indicates that it is easier to find the best design following the model-predicted coefficients (Spearman, 1961). These metrics quantify different aspects of prediction quality: Volume/Surface errors measure field-level discrepancies, $C_D / C_L$ are obtained by integrating the predicted fields over the geometry, and Spearman's $\rho$ evaluates ranking consistency across samples. Accordingly, values that appear identical when rounded to four decimal places in the main text can still differ slightly at higher precision, and small local field improvements need not translate monotonically to integrated coefficients.

# D. Full Experiments Details

## D.1. Model Configurations

**Code Availability.**    Sample code is available at https://github.com/Adversarr/LRSA-Operator.

We employ input/output feature normalization for all tasks to ensure training stability. We use the AdamW optimizer (Loshchilov & Hutter, 2017) coupled with a OneCycle scheduler (Smith & Topin, 2018). Detailed hyperparameters are listed in Table 11. Note that model depth, width, and MLP width are chosen to match the parameter counts of baselines for fair comparison.

## D.2. Statistical Significance

To explicitly quantify the robustness of our method against initialization noise, we trained both LRSA and the strong baseline Transolver on all standard benchmarks using 5 different random seeds. The mean relative $L_2$ error and standard deviation are reported in Table 12. LRSA demonstrates not only lower average error but also comparable or lower variance, indicating stable convergence.

## D.3. Efficiency Benchmark Details

We conduct efficiency benchmarks on a single NVIDIA RTX4090 (48G) GPU using PyTorch 2.9 with CUDA 12.8. LRSA utilizes the native scaled_dot_product_attention, which automatically dispatches to FlashAttention-2 kernels (Dao, 2024) when inputs are in FP16 or BF16. Transolver is implemented using its official open-source codebase. We report the peak memory usage (GB), average forward pass latency (ms), and backward pass latency (ms) averaged over 100 iterations after a warm-up period. The batch sizes are set to ensure the program is bound by GPU: 20 for Navier–Stokes ($N = 64^2$), 20 for Airfoil ($N = 221 \times 51$), and 8 for ShapeNetCar ($N = 32{,}186$).

*Table 11.* Training and model configurations of LRSA. Training configurations are directly from previous works without extra tuning (Bonnet et al., 2022; Hao et al., 2023; Deng et al., 2024). Here $\mathcal{L}_v$ and $\mathcal{L}_s$ represent the mse loss on volume and surface fields respectively. As for Darcy, we adopt an additional spatial gradient regularization term $\mathcal{L}_g$ following ONO (Xiao et al., 2024).

| Problems | Model Configurations | | | | | Training Configurations | | | | |
|---|---|---|---|---|---|---|---|---|---|---|
| | Depth | Width | #Heads | $M$ | #Params | Max Lr | Weight Decay | Epochs | Batch Size | Loss |
| Elasticity | 8 | 128 | 8 | 64 | 2.1M | 1e-3 | 1e-5 | 500 | 1 | |
| Navier Stokes | 8 | 256 | 8 | 32 | 12.0M | 1e-3 | 1e-5 | 500 | 2 | Relative |
| Plasticity | 8 | 128 | 8 | 64 | 2.8M | 1e-3 | 1e-5 | 500 | 8 | L2 |
| Airfoil | 8 | 128 | 4 | 64 | 2.8M | 1e-3 | 1e-5 | 500 | 4 | |
| Pipe | 8 | 128 | 4 | 32 | 2.8M | 1e-3 | 1e-5 | 500 | 4 | |
| Darcy | 8 | 128 | 8 | 64 | 2.8M | 1e-3 | 1e-5 | 500 | 4 | $\mathcal{L}_{rL2} + 0.1\mathcal{L}_g$ |
| Irregular Darcy | 4 | 64 | 4 | 32 | 363K | 1e-3 | 1e-5 | 2000 | 16 | |
| Pipe Turbulence | 4 | 64 | 4 | 32 | 363K | 1e-3 | 1e-5 | 2000 | 16 | Relative |
| Heat Transfer | 4 | 64 | 4 | 32 | 363K | 1e-3 | 1e-5 | 2000 | 16 | L2 |
| Composite | 4 | 64 | 4 | 32 | 363K | 1e-3 | 1e-5 | 2000 | 16 | |
| ShapeNet Car | 8 | 256 | 8 | 64 | 9.1M | 1e-3 | 1e-5 | 400 | 1 | $\mathcal{L}_v + 0.5\mathcal{L}_s$ |
| AirfRANS | 8 | 256 | 8 | 64 | 9.1M | 3e-4 | 1e-5 | 400 | 1 | $\mathcal{L}_v + \mathcal{L}_s$ |

| Dataset | Airfoil | Pipe | Plasticity | Navier-Stokes | Darcy | Elasticity |
|---|---|---|---|---|---|---|
| Transolver | 0.53±0.01 | 0.33±0.02 | 0.12±0.01 | 9.00±0.13 | 0.57±0.01 | 0.64±0.02 |
| Ours | 0.42±0.03 | 0.23±0.01 | 0.044 ± 0.001 | 4.44 ± 0.09 | 0.48 ± 0.01 | 0.33 ± 0.01 |

*Table 12.* Caption ($\times 10^{-2}$)

**Analysis of Latency and Precision.** Detailed results are presented in Table 13. In FP32 precision, LRSA offers comparable forward latency to Transolver. The primary efficiency gain of LRSA stems from its **half-precision stability**. Transolver involves explicit re-normalization and aggregation of physics-aware tokens ($\text{softmax}(\cdot)/\sum \text{softmax}(\cdot)$), an operation sensitive to the reduced dynamic range of FP16/BF16, leading to the instability (divergence) observed. Conversely, LRSA's standard attention formulation integrates normalization within the fused kernel, preserving numerical stability. In FP32, FlashAttention is not available in our setup, so PyTorch SDPA falls back to non-Flash kernels (cuDNN SDPA or the math implementation). Once we switch to FP16/BF16, SDPA dispatches to FlashAttention, and SDPA becomes stable and efficient with no such fallback-related issues.

*Table 13.* **Detailed Efficiency Breakdown.** Comparison of Peak Memory (Mem, in GB), Forward Latency (Fwd, in ms), and Backward Latency (Bwd, in ms) across different precisions. Entries marked with † indicate configurations that resulted in training divergence (NaN loss) or severe accuracy degradation.

| Method | Navier–Stokes | | | Airfoil | | | ShapeNet Car | | |
|---|---|---|---|---|---|---|---|---|---|
| | Mem | Fwd | Bwd | Mem | Fwd | Bwd | Mem | Fwd | Bwd |
| *Configurations* | $N = 4096, M = 64, B = 20$ | | | $N = 11271, M = 64, B = 20$ | | | $N = 32186, M = 64, B = 8$ | | |
| Transolver-FP32 | 8.7 | 70.2 | 129.3 | 18.4 | 127.9 | 253.0 | 28.2 | 163.3 | 273.3 |
| Transolver-BF16† | 6.9 | 43.2 | 85.3 | 17.1 | 84.7 | 214.3 | 20.5 | 88.2 | 206.4 |
| Transolver-FP16† | 6.9 | 43.4 | 85.2 | 17.1 | 82.2 | 214.3 | 20.5 | 88.1 | 206.5 |
| Ours-FP32 | 10.5 | 48.0 | 98.1 | 12.1 | 53.2 | 178.4 | 26.2 | 103.3 | 238.4 |
| Ours-BF16 | 5.5 | 24.5 | 51.8 | 6.5 | 27.8 | 76.6 | 13.3 | 40.2 | 97.0 |
| Ours-FP16 | **5.5** | **21.5** | **37.9** | **6.5** | **20.0** | **58.5** | **13.3** | **40.2** | **96.9** |

## D.4. Extended Ablation Results

We provide the complete numerical results for the ablation studies discussed in the main text, including component analysis, rank sensitivity, and training precision impact (Table 14).

*Table 14.* **Full Ablation Results.** Relative $L_2$ errors across standard benchmarks under various ablation settings.

| Variants \ Dataset | Airfoil | Pipe | Plasticity | Navier-Stokes | Darcy | Elasticity |
|---|---|---|---|---|---|---|
| *Ablation: Components* | | | | | | |
| w/o Intra Attention | 4.30e-3 | 2.60e-3 | 7.09e-4 | 5.44e-2 | 4.46e-3 | 3.60e-3 |
| Enforce Symmetric | 4.60e-3 | 2.50e-3 | 4.71e-4 | 5.93e-2 | 4.63e-3 | 4.70e-3 |
| *Ablation: Latent Rank M* | | | | | | |
| 8 | 4.32e-3 | 3.01e-3 | 4.87e-4 | 1.03e-2 | 4.85e-3 | 3.71e-3 |
| 16 | 4.47e-3 | 2.32e-3 | 4.74e-4 | 9.10e-2 | 4.75e-3 | 3.47e-3 |
| 32 | 4.31e-3 | **2.06e-3** | 4.73e-4 | 6.91e-2 | **4.31e-3** | 3.57e-3 |
| 64 | 4.23e-3 | 2.67e-3 | 4.65e-4 | 4.84e-2 | 4.52e-3 | **3.32e-3** |
| 128 | **4.15e-3** | 2.75e-3 | 4.45e-4 | 4.32e-2 | 4.16e-3 | 3.40e-3 |
| 256 | 4.86e-3 | 2.98e-3 | **4.43e-4** | **3.19e-2** | 4.28e-3 | 3.41e-3 |
| *Precision (Training & Evaluation)* | | | | | | |
| Float 32 | 4.31e-3 | 2.24e-3 | 4.65e-4 | 4.84e-2 | 4.31e-3 | 3.32e-3 |
| Float 16 | 4.45e-3 | 2.49e-3 | 4.73e-4 | 4.85e-2 | 4.32e-3 | 3.35e-3 |
| BFloat 16 | 4.50e-3 | 2.89e-3 | 5.13e-4 | 5.12e-2 | 4.30e-3 | 3.58e-3 |

