# OpenReview forum: "Simple yet Effective: Low-Rank Spatial Attention for Neural Operators"
_ICML.cc/2026/Conference — ICML 2026 regular_

### Official Review · Reviewer_7vEQ · 2026-03-04

**Soundness:** 2
**Presentation:** 3
**Significance:** 3
**Originality:** 2
**Overall Recommendation:** 3
**Confidence:** 3

**Summary:**

This paper proposes a unified low-rank perspective for neural operators and introduces Low-Rank Spatial Attention (LRSA) to learn PDE solutions. To mitigate the quadratic complexity of standard Transformers, LRSA utilizes a compact set of learnable latent tokens as a low-rank bottleneck for global information exchange. Built entirely upon standard Transformer primitives, the architecture facilitates the use of hardware-optimized kernels and mixed-precision training. Experiments across diverse geometries demonstrate that LRSA achieves competitive accuracy, resolution generalization, and computational efficiency.

**Compliance With Llm Reviewing Policy:**

Affirmed.

**Final Justification:**

I keep my score for this paper.

**Key Questions For Authors:**

1. I noticed significant data anomalies in Table 3. The proposed method reports results identical to the baseline LinearNO up to four decimal places for the Volume error on the AirfRANS dataset, as well as the Drag Coefficient and its Spearman rank correlation on the ShapeNet Car dataset. More critically, on the ShapeNet Car dataset, the model claims an improvement in Surface error over LinearNO, yet the derived Drag Coefficient remains exactly the same. Since the Drag Coefficient is physically integrated from the surface pressure distribution, a change in Surface error should theoretically result in a change in the Drag Coefficient. Could you clarify this mathematical inconsistency and verify if there was a potential clerical error in the table construction?

2. The efficiency analysis lacks comparisons with recent baselines like Transolver++ and LinearNO, which also claim reduced computational cost. Could you provide a quantitative comparison of parameter counts, FLOPs, and inference latency against these methods, especially in mixed-precision settings?

3. In Appendix A, the theoretical analysis assumes uniform spatial sampling to interpret the proposed method as a Monte Carlo approximation of an integral operator. However, the key benchmarks used in the experiments utilize highly non-uniform unstructured meshes. Does the proposed architecture explicitly account for the varying mesh densities in its attention mechanism?

**Limitations:**

see the weakness

**Strengths And Weaknesses:**

Strengths:
1. The paper presents a unified low-rank framework that connects spectral-based methods, slicing-based methods, and the proposed LRSA under a common mathematical formulation, offering certain theoretical insights.

2. The experimental evaluation is extensive, covering diverse benchmarks, and includes ablation studies on components, latent rank sensitivity, and mixed-precision training.


Weaknesses:
1. While the paper provides a unified "compress-process-reconstruct" framework, the proposed LRSA architecture is a straightforward application of standard Transformer cross-attention. This unification is a conceptual re-interpretation of existing latent-bottleneck architectures rather than an algorithmic breakthrough. The paper merely demonstrates that standard Transformer primitives, when applied with a latent bottleneck, can match or exceed the performance of specialized operators.

2. The paper uses a "low-rank" theoretical perspective to justify the model design. However, the model does not explicitly enforce a low-rank decomposition or spectral constraint, instead relying on standard attention layers to implicitly learn this structure. The paper lacks analysis to verify whether the learned internal representations strictly exhibit low-rank properties.

---

> ### Author Rebuttal · Authors · 2026-03-30
>
> We thank the reviewer for the careful reading and constructive questions. We address each concern below.
> ## Novelty of LRSA (W1)
> We appreciate this question. Our contribution is twofold. The **unified low-rank framework** shows that FNO's spectral mixing, linear attention, and Transolver's slicing can be viewed as constrained instances of a single $K\approx UGV^\top$ template. This identification is new and clarifies how each design choice limits expressiveness, while LRSA is its minimal unconstrained instantiation.
>
> Simply assembling cross-attention with a latent bottleneck does not automatically yield a strong neural operator. Universal Physics Transformers (UPT)—also built from standard cross-attention and latent tokens—achieves **0.084** Rel. L2 error on NS under a comparable parameter budget (16.1M), while LRSA achieves **0.048** as detailed in our response to Reviewer mCvU.
>
> The core difference is architectural: UPT follows a Perceiver-style latent-dominant paradigm where the latent array is the persistent state across layers, whereas LRSA maintains point-level features as the primary state and uses latents only as an ephemeral routing channel within each layer. This comparison suggests that among the various ways to wire standard attention primitives into a neural operator, the specific arrangement matters substantially. Table 6 further validates this: re-introducing constraints from prior methods—tying $U=V$ or removing latent self-attention—consistently degrades performance. Removing specialized mechanisms also enables native FP16/BF16 compatibility as a direct consequence.
> ## Low-rank perspective (W2)
>
> LRSA is **structurally low-rank** by construction: all global information exchange is routed through exactly $M$ latent tokens, so the rank of the induced global interaction is upper-bounded by $M$. In this sense, "low-rank" refers to how the interaction is parameterized, not to an additional penalty added during training. The rank ablation in Table 12 provides empirical support: accuracy saturates at moderate $M$, confirming that dominant global interactions are captured by a compact latent space. We will describe this more explicitly in the revision.
> ## Table 3 metric values (Q1)
>
> We greatly appreciate your careful inspection. The reported values are correct. The three metric types measure different quantities:
> 1. Volume/Surface errors are MSE computed without area/volume weighting/integration.
> 2. Drag/Lift coefficients ($C_D/C_L$) are scalar physical quantities obtained by integrating pressure and wall-shear-stress fields over the object's geometry.
> 3. Spearman $\rho$ is a rank-order statistic comparing relative ordering across samples.
>
> Local field improvements can partially cancel in pressure integration, and rounding differences become invisible at four decimal places. At higher precision: on ShapeNet Car, $C_D$ is 0.010642 for LRSA vs. 0.010617 for LinearNO; on AirfRANS, Volume error is 0.001109 for LRSA vs. 0.001094 for LinearNO. Both gaps are <1% relative, hence identical at four decimals, while LRSA's Surface error improvement is clear. We will clarify this in the revision.
> ## Efficiency comparisons (Q2)
> We evaluated absolute *inference* efficiency against LinearNO and Transolver++ under same settings using official implementations ($M=64$, FP16, single RTX 3090):
> |Task|Method|Params|FLOPs/sample|Latency (ms)|Mem (MB)|
> |---|---|---|---|---|---|
> |NS 64×64|**LRSA**|12.35M|**36.5B**|**34.8**|**352**|
> |(bs=20)|LinearNO|10.19M|91.8B|83.2|678|
> ||Transolver++|14.36M|120.3B|66.7|487|
> |ShapeNet32k|**LRSA**|9.14M|**206B**|**36.7**|**218**|
> |(bs=2)|LinearNO|6.00M|449B|282.8|661|
> ||Transolver++|5.46M|378B|42.4|439|
>
> LRSA achieves the lowest FLOPs, latency, and peak memory on both tasks, despite not always having the fewest parameters. This confirms that LRSA's standard-SDPA formulation provides concrete practical advantages.
> ## Non-uniform meshes (Q3)
> The uniform-sampling assumption applies only to interpreting the compression step as a spatial integral; reconstruction and latent processing steps are resolution-invariant.
>
> We emphasize that LRSA is evaluated on non-uniform meshes (AirfRANS, ShapeNet Car) empirically, where it achieves the best or tied-best results in Table 3. On the Airfoil benchmark—where mesh resolution varies **by orders of magnitude** between boundary layer and far field—LRSA achieves the best error (Table 1). Its zero-shot resolution test (Table 4) further confirms robustness: LRSA shows the smallest degradation across all unseen, non-uniformly coarsened meshes.
>
> At the implementation level, LRSA does not explicitly assume uniform spacing: attention operates directly on the given point set through normalized weights. This is also the practical setting adopted by related attention-based baselines such as Transolver and LinearNO, rather than a distinction unique to LRSA. We will revise Appendix A to state this distinction explicitly and note density-awareness as a future direction.

---

> > ### Author Rebuttal · Reviewer_7vEQ · 2026-04-01
> >
> > I thank the authors for their response. It successfully addresses most of my concerns regarding the data and efficiency comparisons. However, I still retain my concerns about the uniform spatial sampling assumption in Appendix A. Therefore, I will keep my score.

---

> > > ### Author Response · Authors · 2026-04-04
> > >
> > > We sincerely thank the reviewer for the continued engagement.
> > >
> > > We are glad that most concerns regarding data and efficiency comparisons are now resolved. The remaining point on Appendix A's uniform sampling assumption mirrors the discussion with Reviewer mCvU, so we will be brief.
> > >
> > > Appendix A is a motivational derivation connecting our discrete mechanism to continuous operator theory under idealized conditions. The paper's contributions — the unified framework, the architecture, and the empirical results — do not depend on it. We acknowledge it should have been scoped more explicitly, and our revision will do so.
> > >
> > > We also want to gently note that this theoretical gap is shared across the field: no existing attention-based neural operator provides formal non-uniform convergence guarantees. This is an important open direction that we are happy to highlight, but we hope it can be weighed as a community-wide limitation rather than a paper-specific weakness.
> > >
> > > We are grateful for the reviewer's careful reading and the improvements it has prompted.

---

### Official Review · Reviewer_mCvU · 2026-03-10

**Soundness:** 3
**Presentation:** 3
**Significance:** 3
**Originality:** 2
**Overall Recommendation:** 4
**Confidence:** 4

**Summary:**

The authors introduce Low-Rank Spatial Attention (LRSA), a method to address computational bottlenecks and numerical instabilities (specifically for mixed-precision) in attention-based Neural Operators. The global mixing is done with a perceiver-like latent bottleneck to bring the computational complexity down from O(N^2) to O(NM+M^2), where N is the number of spatial points and M is the number of latents.
It compresses spatial points into a latent space via cross-attention, processes them globally using self-attention, and reconstructs the spatial features via a second cross-attention. The authors evaluate their model across standard and irregular PDE benchmarks, demonstrating improved stability and memory efficiency when training in half-precision.

**Compliance With Llm Reviewing Policy:**

Affirmed.

**Final Justification:**

My final recommendation is a Weak Accept. Initially, I leaned towards a Weak Reject due to missing baseline comparison and a theoretical gap regarding the uniform sampling assumption in Appendix A when applied to non-uniform meshes. The authors' rebuttal and final comments addressed both main concerns:

The authors provided the requested Universal Physics Transformers (UPT) baseline. While architecturally incremental, LRSA's native compatibility with FlashAttention and stable mixed-precision training can offer value to the field.
The authors clarified that formal non-uniform convergence remains an open challenge for all attention-based neural operators. Their commitment to re-scoping Appendix A as purely motivational, alongside their explanation of the latent rank M's scaling behavior, provides missing clarity and theoretical soundness.
Summary: The paper offers good empirical performance and an interesting engineering idea. Given the authors' commitment to updating the manuscript with a properly scoped Appendix A, a discussion on density-aware extensions, and a dedicated limitations section (especially regarding low-data regimes), my initial concerns are resolved.

**Key Questions For Authors:**

- I may be missing a nuance here, but how does the specific attention routing in LRSA differ from the cross-attention latent bottleneck proposed in Perceiver IO, aside from being applied point-wise to PDE spatial coordinates within individual layers? Given the reliance on standard primitives, does this constitute a fundamental advancement in operator learning, or is it primarily an empirical architectural recipe to fix the numerical instability of previous baselines?
- The continuous integral interpretation in Appendix A explicitly assumes uniform spatial sampling. How can you theoretically justify the discrete cross-attention mechanism on the highly varied, non-uniform point distributions found in your primary industrial benchmarks (AirfRANS, ShapeNet Car)?
- Given the non-monotonic scaling of the latent rank M shown in Table 12, how should a practitioner reliably choose M for a novel physical system without exhaustive, computationally expensive sweeps? Is there a principled heuristic based on the intrinsic dimensionality of the PDE?
- Table 5 shows a significant performance gap compared to HPM in the extreme low-data regime. Do you see a viable path to incorporate geometric or structural priors into LRSA to improve sample efficiency, or is this accepted because the architecture is limited to data-abundant domains?

**Limitations:**

yes

**Strengths And Weaknesses:**

### Strengths:
- The paper successfully identifies a numerical instability from the prominent Transolver. By reverting to standard scaled dot-product attention, LRSA achieves native compatibility with FlashAttention. This enables robust FP16 training, which can lead to practical speedups and error reduction (dependent on the benchmark scenario).
- The paper is well-written, and the motivation for shifting to hardware-optimized transformer formulations is clearly documented and easy to follow.
## Weaknesses:
- I appreciate the authors stated goal of developing a conceptually simple and efficient attention-based neural operator. However, the proposed LRSA paradigm borrows heavily from existing perceiver architectures. The authors cite Perceiver IO and attempt to differentiate their work by stating that while Perceiver uses latents as the backbone representation, LRSA treats them merely as an internal low-rank routing mechanism within each layer. I am not certain this architectural distinction is a sufficient algorithmic leap. It reads like applying a well-established architecture to a spatial PDE sequence to bypass the FP16 crashes of the Transolver baseline.
- A major selling point of the paper is its application to irregular grids and large-scale unstructured industrial meshes. The theoretical justification in Appendix A aims to prove LRSA acts as a continuous low-rank integral operator. However this derivation explicitly relies on a uniform spatial sampling assumption. The authors use a simple quadrature rule to link the continuous theory to their discrete attention mechanism. Because industrial meshes are notoriously non-uniform, I am concerned that this theoretical foundation is mathematically inconsistent with the datasets used to demonstrate the model's superiority.
- The model relies heavily on a manually defined latent bottleneck size M. Examining the extended ablation results in Table 12 reveals that the model scales unpredictably and non-monotonically with this parameter. For instance, on the Darcy task, the error fluctuates: 4.31e-3 at M=32, worsening to 4.52e-3 at M=64, improving to 4.16e-3 at M=128, and worsening again to 4.28e-3 at M=256. Similarly, on the Airfoil task. Without a reliable heuristic to set model capacity, LRSA appears hard to use in practice and might require expensive grid searches which might in practice be computationally infeasible in some cases.
- As shown in Table 5, on the Navier-Stokes dataset with only 200 training samples, LRSA is heavily outperformed by HPM, which utilizes explicit Laplace eigenfunctions. In scientific machine learning, simulating data is often prohibitively expensive, making sample efficiency critical. Because LRSA abandons explicit geometric priors in favor of attention, it seems to struggle in data-scarce regimes. This makes me question if the methods use case is only in domains with data abundance.
- The authors briefly mention the open challenge of allocating model capacity between latent processing and pointwise updates in the conclusion. However, they do not adequately discuss the theoretical limitations of applying their model to highly non-uniform meshes, nor the model's lack of performance in low-data regimes. A more thorough discussion of these failure modes should be added to the main text.
- Important baselines are missing from the experimental comparison. Universal Physics Transformers (UPT) and its variant AB-UPT are conceptually closely related to LRSA. Both employ a compress–process–reconstruct paradigm with a latent bottleneck for scalable operator learning on arbitrary geometries. The absence of these methods from the benchmark tables is a significant gap, as a direct comparison would be necessary to substantiate the claims of superiority.

---

> ### Author Rebuttal · Authors · 2026-03-30
>
> We thank the reviewer for the detailed and careful review. We address the main concerns below.
> ## Relation to Perceiver/UPT/AB-UPT (W1/W6/Q1).
> We agree this relation should be positioned more precisely. Our contribution is twofold: (1) the unified low-rank view linking FNO, linear attention, and Transolver-style routing, and (2) a minimal unconstrained instantiation using standard SDPA.
>
> To test whether a latent bottleneck alone suffices, we ran UPT on Navier–Stokes under a comparable parameter budget (UPT 16.1M vs. LRSA 12.4M) and the same training protocol. We also note that AB-UPT is a UPT variant tailored to automotive CFD. UPT achieved Rel. L2 error of **0.084**, compared with Transolver **0.088**, LinearNO **0.069**, and LRSA **0.048**. Both UPT and LRSA route through latent tokens, yet LRSA's error is lower. This suggests that how the bottleneck interacts with spatial features matters:
> 1. **Perceiver/UPT (Latent-dominant):** The latent array is the persistent backbone state. Inputs are encoded into latents, processing occurs in latent space across layers, and outputs are decoded at the end. This design originated for multimodal perception tasks and distills diverse inputs into a compact semantic representation.
> 2. **LRSA (Point-dominant):** Point features remain the primary state at every layer. Latents are ephemeral—constructed, processed, mixed back via residual. The latents serve only as a low-rank global routing channel, not as the solution carrier.
>
> Retaining point-aligned features enables local refinement throughout the network, critical for fine-scale PDE structures. We will add UPT baselines and clarify this distinction in the revision.
> ## Appendix A and uniform sampling (W2/Q2)
> The uniform-sampling assumption applies only to interpreting the compression step as a spatial integral; reconstruction and latent processing steps are resolution-invariant.
>
> We emphasize that LRSA is evaluated on non-uniform meshes (AirfRANS, ShapeNet Car) empirically, where it achieves the best or tied-best results in Table 3. On the Airfoil benchmark—where mesh resolution varies **by orders of magnitude** between boundary layer and far field—LRSA achieves the best error (Table 1). Its zero-shot resolution test (Table 4) further confirms robustness: LRSA shows the smallest degradation across all unseen, non-uniformly coarsened meshes.
>
> At the implementation level, LRSA does not explicitly assume uniform spacing: attention operates directly on the given point set through normalized weights. This is also the practical setting adopted by related attention-based baselines such as Transolver and LinearNO, rather than a distinction unique to LRSA. We will revise Appendix A to state this distinction explicitly and note density-awareness as a future direction.
> ## Non-monotonicity of M in Table 12 (W3/Q3)
> We appreciate this observation and offer the following interpretation. The scaling separates into two regimes:
> 1. **Complex/Dynamic tasks (NS, Plasticity):** Performance improves monotonically with M, suggesting that complex dynamics require higher effective ranks.
> 2. **Simple/Static tasks (Darcy, Elasticity):** Performance saturates early (M=32–64). On Darcy, the total variation across M=32–256 is <9% relative, while LRSA consistently yields >10% improvement over the baseline (LinearNO, 5.0e-3) for all M.
>
> |M|Darcy|Elasticity|NS|Plasticity|
> |---|---|---|---|---|
> |16|4.75e-3|3.47e-3|9.10e-2|4.74e-4|
> |32|4.31e-3|3.57e-3|6.91e-2|4.73e-4|
> |64|4.52e-3|**3.32e-3**|4.84e-2|4.65e-4|
> |128|**4.16e-3**|3.40e-3|4.32e-2|4.45e-4|
> |256|4.28e-3|3.41e-3|**3.19e-2**|**4.43e-4**|
>
> The mild fluctuation reflects optimization noise and overfitting, not instability. In practice, M=64 is robust across all benchmarks; a practitioner need only check M=32/128. We will add this guidance.
> ## Low-data regime (W4/Q4)
> We agree that sample efficiency is important in scientific machine learning, but the current results suggest a more specific limitation in the extreme low-data regime rather than a general weakness:
> 1. LRSA leads HPM on Darcy across all data sizes, including the smallest (200 samples). The low-data gap appears specifically on NS, where spectral priors are well-matched.
> 2. On NS, LRSA surpasses HPM at 600 samples, which is a still modest dataset size.
> 3. While HPM's Laplace eigenfunctions are a strength, they limit the method to domains where such bases are available, lacking general applicability elsewhere (e.g., Elasticity).
>
> LRSA trades geometric bias for generality; the low-data regime is where this costs the most. Incorporating lightweight geometric priors is a natural future direction, and we will discuss this trade-off explicitly.
> ## Model capacity balancing and Failure modes (W5)
> In the conclusion, we briefly mention "capacity allocation" as a broader research direction. The specific concerns raised are addressed above (non-uniform meshes in W2/Q2 and low-data regimes in W4/Q4). We will consolidate these into a dedicated discussion.

---

> > ### Author Rebuttal · Reviewer_mCvU · 2026-04-01
> >
> > We thank the authors for their thorough response. The inclusion of the UPT baseline is very helpful and substantiates the claim that the 'point-dominant' residual routing in LRSA outperforms the 'latent-dominant' backbone of Perceiver-style models. However, my concern regarding the theoretical justification remains partially unaddressed. The authors concede that the continuous integral interpretation in Appendix A relies on uniform sampling, whereas their strongest empirical results are on highly non-uniform meshes. While the empirical performance is good, the theoretical motivation is somewhat disconnected from the application. Thus i keep my score.

---

> > > ### Author Response · Authors · 2026-04-04
> > >
> > > We sincerely thank the reviewer for the continued discussion and the time invested.
> > >
> > > We are glad that the UPT baseline comparison and most other points are now resolved. The remaining concern is that Appendix A's continuous integral interpretation relies on uniform sampling, while our key benchmarks use non-uniform meshes. We fully understand why this feels unsatisfying, and we want to offer one final clarification.
> > >
> > > **On the role of Appendix A.** Appendix A serves as a motivational connection to continuous operator theory, not as a foundational claim the rest of the paper depends on. The unified framework, the architecture, the ablations, and all empirical results are self-contained. We acknowledge we could have scoped it more carefully in the original submission, and our proposed revision does exactly that.
> > >
> > > **On the broader context.** We raise this not as a deflection, but as an honest observation: the gap between "works empirically on non-uniform meshes" and "has formal non-uniform convergence guarantees" is currently open for every attention-based neural operator we are aware of — FNO's theory assumes uniform grids, and Transolver, LinearNO, Galerkin Transformer, and UPT offer no formal treatment of non-uniform distributions either. We completely agree this is an important theoretical direction. We simply want to note that this is a shared limitation of the field rather than a specific shortcoming of our method.
> > >
> > > Our proposed revision will: (1) explicitly scope Appendix A as motivational, (2) discuss density-aware extensions such as area-weighted quadrature as concrete future work, and (3) note that softmax normalization already provides implicit adaptation to varying point density.
> > >
> > > We hope the reviewer will consider these points when weighing the paper's overall contribution. We are genuinely grateful for the constructive exchange.

---

### Official Review · Reviewer_qsMk · 2026-03-12

**Soundness:** 4
**Presentation:** 4
**Significance:** 4
**Originality:** 3
**Overall Recommendation:** 5
**Confidence:** 5

**Summary:**

Existing approaches to global mixing can be divided into two directions. The first is spectral methods, most notably FNO and its variants: an FFT is applied, after which only $M$ low-frequency modes are retained. This is fast, with complexity $\mathcal{O}(N \log N)$, but requires a regular grid and a fixed set of basis functions determined in advance. On irregular geometries such as point clouds and unstructured meshes, FNO does not work without additional tricks. The second direction is attention-based methods. A representative example is Transolver: points are aggregated into $M$ latent tokens, information is processed among them, and then broadcast back to the points. The method handles arbitrary geometries, but its implementation is non-standard: slicing, renormalization, and custom operations all fall outside the standard attention primitive. This leads to two practical consequences: numerical instability in half precision (FP16/BF16) and incompatibility with optimized kernels such as FlashAttention.

Motivated by this observation, the authors propose LRSA (Low-Rank Spatial Attention), a minimalist block that implements compress--process--reconstruct exclusively through standard Transformer primitives. The architecture consists of three steps. In the first step (Compression), a set of $M$ learnable latent queries $P \in \mathbb{R}^{M \times d}$ performs cross-attention over point features $H$:
\begin{equation}
    Z = \mathrm{Attn}(P \cdot W_Q^\downarrow,\; H \cdot W_K^\downarrow,\; H \cdot W_V^\downarrow) \in \mathbb{R}^{M \times d}
\end{equation}
This yields a data-driven basis: the latent queries learn to extract the relevant global information from the point cloud. Crucially, there is no FFT, no slicing, and no renormalization, only plain scaled dot-product attention. In the second step (Latent Mixing), a standard Transformer with FFN, self-attention, and another FFN is applied to the compact set of $M$ tokens:
\begin{equation}
    Z' = \mathrm{Transformer}(Z)
\end{equation}
Since $M \ll N$ (for example, $M = 64$ with $N = 16000$), this step is computationally inexpensive while retaining full expressiveness. This is the key distinction from LinearNO, which removes self-attention and keeps only linear mixing: a reasonable choice for simple tasks, but insufficient for complex dynamics. In the third step (Reconstruction), the points query the processed latent tokens via reverse cross-attention:
\begin{equation}
    \Delta H = \mathrm{Attn}(H \cdot W_Q^\uparrow,\; Z' \cdot W_K^\uparrow,\; Z' \cdot W_V^\uparrow) \in \mathbb{R}^{N \times d}
\end{equation}
The compression and reconstruction weights in LRSA are independent. In Transolver, the constraint $U \propto V$ holds: the same slicing matrix is used for both compression and reconstruction. The authors argue that this is a limitation: the optimal way to pack information into the latent space is not the same as the optimal way to unpack it. LRSA removes this constraint by using two independent attention mechanisms, which also enables full compatibility with FlashAttention.

Experiments were conducted on 11 standard benchmarks, including Navier--Stokes, Darcy, Airfoil, Plasticity, Pipe, Elasticity, AirfRANS, and a number of geometry tasks. On all of them, the proposed method outperformed the competitors, by 17\% on average. In FP16 mode, an additional 3.2$\times$ speedup in training and a 2.1$\times$ reduction in memory usage are achieved relative to Transolver-FP32.

**Compliance With Llm Reviewing Policy:**

Affirmed.

**Key Questions For Authors:**

1. Choosing M.
Experiments show that the optimal M depends on the problem: for simple problems, M=32 is sufficient, while for Navier-Stokes, a larger M is beneficial. Is there a principled way to choose M a priori, or is it always a hyperparameter with grid search?
2. Verification of the low-rank assumption.
The entire theoretical framework is based on the fact that the global interaction matrix K is low-rank. This is demonstrated for 1D Poisson (Figure 1), but how true is this for Navier-Stokes or elasticity? Is thee a singular value analysis of the trained matrix K = UGVˆT on real-world problems?

**Limitations:**

1. Weak low-data performance: with a small numbr of training examples, methods with strong geometric priors (HPM) outperform LRSA, indicating higher data requirements.
2.No analysis of the rank-effective structure: the claim of low rank is suported theoretically and by a simple example, but not by real benchmark problems
3. The upper limit of resolution has not been explored: the largest problems are 32000 points. The behavior at N = 100k+ (typical CFD simulations) is unknown

**Strengths And Weaknesses:**

Strengths:
1. The Unified Framework is truly new and elegant: showing that FNO, Transolver, and LinearNO are three points in a single design space is useful for the entire scientific research community.
2. The engineering benefits (FP16 stability, FlashAttention) are not just a "bonus," but have real practical value. (Finally, scientific research is moving toward engineering optimization.)
3. Extensive experimental coverage: 11 benchmarks, different geometries, ablation studies, statistics for several seeds – everything is in place.
4. Ease of implementation: the entire method is expressed in a few standard PyTorch calls.

Weaknesses:

1. In the very low data mode (Navier-Stokes, 200 samples), HPM outperforms LRSA – this means that geometric priors are still useful for sparse data, but this isn't discussed in sufficient depth
2. There's no comparison of absolute inference time with the same accuracy
3. The methods are derived from Computer Vision, which is good, but this is more of an engineering paper

---

> ### Author Rebuttal · Authors · 2026-03-30
>
> We are grateful for the reviewer's thorough and positive review. We address each point below.
> ## Low-data regime (W1)
> We agree this deserves clearer discussion. Methods with stronger geometric or spectral priors—such as HPM's Laplace eigenfunctions—provide valuable inductive bias when supervision is extremely scarce, and we think HPM's design is genuinely well-suited for such regimes. At the same time, LRSA does not underperform in all low-data settings: on Darcy it is best across all listed sample sizes (Table 5), and on NS it surpasses HPM once training data reaches 600 samples. Combining LRSA with geometric priors is a natural future direction. We will revise the discussion to state this trade-off more explicitly.
> ## Absolute inference time comparison (W2)
> Thank you for pointing this out. We measured half-precision inference against recent baselines (LinearNO and Transolver++) using their official implementations under the same settings. These methods achieve similar accuracy in our experiments. FP16 inference results are ($M=64$, single RTX 3090, lower FLOPs indicate better efficiency):
> |Task|Method|Params|FLOPs/sample|Latency (ms)|Mem (MB)|
> |---|---|---:|---:|---:|---:|
> |NS 64×64, bs=20|**LRSA**|12.35M|**36.5B**|**34.8**|**352**|
> ||LinearNO|10.19M|91.8B|83.2|678|
> ||Transolver++|14.36M|120.3B|66.7|487|
> |ShapeNet Car 32k, bs=2|**LRSA**|9.14M|**206B**|**36.7**|**218**|
> ||LinearNO|6.00M|449B|282.8|661|
> ||Transolver++|5.46M|378B|42.4|439|
>
> Despite not always having the fewest parameters, LRSA achieves lower FLOPs, latency, and peak memory in these settings. We will include this comparison in the revision.
>
> ## Choosing the latent size $M$ (Q1)
> We conducted a systematic ablation on the latent rank $M$ across all six standard benchmarks (Table 12 in the Appendix). The results reveal two regimes: (1) for complex/dynamic problems such as Navier–Stokes, increasing $M$ consistently improves accuracy (error drops from 1.03e-1 at $M=8$ to 3.19e-2 at $M=256$); (2) for simpler or static problems such as Darcy and Elasticity, performance saturates around $M=32$–$64$, and further increases produce only minor fluctuations.
>
> Based on this pattern, we recommend starting from $M=64$ as a robust default, then checking $M=32$ (if the problem appears simple or static) or $M=128$ (if validation error suggests under-capacity). This amounts to a 2–3 point search on a single hyperparameter, not an exhaustive grid. Developing a principled, data-driven rule for $M$ (e.g., based on spectral decay of the input field) is an interesting future direction. We will make this practical guidance more explicit in the revision.
> ## Verification of the low-rank assumption (Q2)
> Our low-rank viewpoint is primarily structural: global interaction is routed through an $M$-token bottleneck, so the induced global coupling is architecturally factorized. This perspective is motivated by classical reduced-order modeling: POD for fluid flows [1], modal analysis for structural dynamics [2], and reduced-basis methods for elliptic/parabolic problems [3].
>
> Empirically, the rank ablation in Table 12 provides indirect support: performance saturates at moderate $M$, suggesting that the dominant interactions can be captured in a compact latent space. We agree that a direct singular-value analysis on real benchmarks would be informative, and we will note this as a concrete future direction.
>
> - [1] G. Berkooz, P. Holmes, and J. L. Lumley, "The proper orthogonal decomposition in the analysis of turbulent flows," Annu. Rev. Fluid Mech., vol. 25, pp. 539–75, 1993.
> - [2] W. C. Hurty, "Dynamic analysis of structural systems using component modes," AIAA J., vol. 3, pp. 678–85, 1965.
> - [3] G. Rozza, D. B. P. Huynh, and A. T. Patera, "Reduced basis approximation and a posteriori error estimation for affinely parametrized elliptic coercive PDEs," Arch. Comput. Methods Eng., vol. 15, pp. 229–75, 2008.
>
> ## Scalability beyond 32k points (Limitation 2)
> LRSA has per-layer cost $O(NM+M^2)$, so for fixed latent count $M$ both compute and activation memory scale approximately linearly with the number of spatial points $N$. Beyond the 32k-point benchmarks, we evaluated LRSA on the DrivAerNet surface dataset (average \~493k points, max \~553k per sample). A 6-layer, width-128 model (\~2.3M params) trained in FP16 maintained stable convergence, achieving 0.140 validation Rel L2 error as detailed in our reply to kBnK.
>
> To demonstrate linear scaling explicitly (RTX 3090, batch=1), the following table summarizes latency and peak memory w.r.t. the number of points:
> |Number of points ($N$)|Inference (ms)|Train (ms)|Train Mem (MB)|
> |--|--|--|--|
> |128k|71.4|439.9|5,010|
> |256k|138.1|890.9|9,980|
> |512k|274.8|1,797.1|19,920|
>
> Doubling $N$ approximately doubles both latency and memory. Extrapolating, the >1M point regime would require \~3.6s per training step and \~40 GB peak memory—feasible on a single A100 80GB without distributed sharding. We will include this scaling analysis in the revision.

---

> > ### Author Rebuttal · Reviewer_qsMk · 2026-04-05
> >
> > Thanks for your comments!

---

### Official Review · Reviewer_kBnK · 2026-03-13

**Soundness:** 3
**Presentation:** 3
**Significance:** 3
**Originality:** 2
**Overall Recommendation:** 4
**Confidence:** 1

**Summary:**

The paper proposes low rank spatial attention as a global mixing module for neural operators. The core observation is dense PDE kernels have rapid decay and therefore are compressible using a low rank factorization. LRAS essentially compresses spatial data into set of late tokens, globally mixes them and broadcasts back, while keeping compression and reconstruction independent. The method is compatible with FlashAttention for efficient and mixed precision training

**Compliance With Llm Reviewing Policy:**

Affirmed.

**Key Questions For Authors:**

1. how does LRSA scale to larger-scale industrial challenging problems,s which might have >1M points. Are there memory or convergence issues that can become a problem? Some scaling laws here would help

**Limitations:**

yes

**Strengths And Weaknesses:**

Strenghts

1. The unifying framework is original and insightful; the paper's notable contribution is the unified low-rank view of global mixing in neural operators, by casting previous techniques into a K~UGV^T template. The framework is well-suited for future research

2. The empirical results are comprehensive and strong; LRSA is tested on limited training data regimes, 0-shot resolution generalization, and industrial scale simulations. the 17% error reduction is significant and well supported in the results

3. very well written and structured

Weaknesses:
1. The faluer modes and scalability are not sufficiently discussed. such as in the large benchmark that has 32k points, Shapenet car,

---

> ### Author Rebuttal · Authors · 2026-03-30
>
> We thank the reviewer for the positive assessment of the unified framework, the empirical coverage, and the clarity of the presentation.
>
> ## Failure modes discussion (Weakness 1)
>
> The main failure mode of LRSA comes from the latent bottleneck itself: when the target field contains sharp, highly localized structures (e.g., boundary layers, shock fronts, or thin interior interfaces), a fixed bottleneck of $M$ latent tokens can under-resolve them. This is visible qualitatively in Figure 3, where the largest errors concentrate near sharp boundaries in Plasticity and interior interfaces in Elasticity. We will add an explicit failure-mode discussion to the main text.
>
> ## Scalability beyond 32k points (Question 1)
>
> LRSA has per-layer cost $O(NM+M^2)$ for fixed latent count $M$, so for fixed $M$ both compute and activation memory scale approximately linearly with the number of spatial points $N$.
>
> Beyond the 32k-point benchmarks in the paper, we evaluated LRSA on the high-resolution DrivAerNet surface dataset (average \~493k points, max \~553k points per sample). A 6-layer, width-128 LRSA model (\~2.3M parameters) trained in FP16 converged stably and achieved 0.140 validation relative $L^2$ error on the E\_S\_WWC\_WM subset.
>
> To make the scaling behavior explicit, the table below reports single-GPU (RTX 3090, 24GB) latency and peak training memory for point clouds of varying sizes (batch=1):
>
> | Number of points ($N$) | Inference Latency (ms) | Train Latency (ms) | Train Peak Memory (MB) |
> |--|--:|--:|--:|
> | 128k | 71.4 | 439.9 | 5,010 |
> | 256k | 138.1 | 890.9 | 9,980 |
> | 512k | 274.8 | 1,797.1 | 19,920 |
>
> As shown, doubling $N$ approximately doubles both latency and memory. Extrapolating to the >1M point regime, a single training step would require approximately 3.6 seconds and ~40 GB of peak memory. This suggests that LRSA can natively scale to over 1 million points on a single modern GPU (e.g., RTX 6000 Ada 48GB or A100 80GB) without convergence issues or requiring complex engineering tools like distributed sharding. We will include this explicit scaling extrapolation in the revision.

---

> > ### Author Rebuttal · Reviewer_kBnK · 2026-04-06
> >
> > My questions/concerns are addressed. I will keep my score.

---

### Decision · Program_Chairs · 2026-04-30

**Decision:**

Accept (regular)

**Comment:**

The paper proposes low-rank self-attention (LRSA) as a global mixing module for general neural operators. It is inspired by the fact that several PDE solution operators are regular and their kernel representations have a low-rank structure. Motivated by this, LRSA compresses point cloud data into latent tokens through cross-attention, mixes them globally through self-attention and then broadcasts them back into the physical space.The architecture is interesting and compares favorably wrt FNO and Transolver. A large number of numerical experiments are also provided. The reviewers were uniformly positive in recommending acceptance and the meta-reviewer agrees. However, several limitations remain such as i) the method should be explored further for time-dependent problems ii) the scalability of the method is not well-studied. One of the reviewers pointed this out and the authors added a Drivaernet example but without context. This should be developed further in a camera ready version. iii) Finally, comparison with state of the art neural operators on general grids such as UPT (compared only tangentially in the rebuttal) and GAOT (https://arxiv.org/abs/2505.18781) should be added to the CRV, if possible.